# A Lagrangian Analysis of Pockets of Open Cells over the Southeast Pacific

Kevin M. Smalley[1], Matthew D. Lebsock[1], Ryan Eastman[2], Mark Smalley[1,3], and Mikael K. Witte[1,3,4]

[1]Jet Propulsion Laboratory, California Institute of Technology, Pasadena, California, USA
[2]Department of Atmospheric Sciences, University of Washington, Seattle, Washington, USA
[3]Joint Institute for Regional Earth System Science and Engineering, University of California, Los Angeles, California, USA
[4]Naval Postgraduate School, Meteorology, Monterey, California, USA

**Correspondence:** Kevin M. Smalley (ksmalley@jpl.nasa.gov)

**Abstract.**

Pockets of open cells (POCs) have been shown to develop within closed-cell stratocumulus (StCu) and a large body of evidence suggests that the development of POCs result from changes in small-scale processes internal to the boundary layer rather than large-scale forcings. Precipitation is widely viewed as a key process important to POC development and main-
tenance. In this study, GOES-16 satellite observations are used in conjunction with MERRA-2 winds to track and compare the microphysical and environmental evolution of two populations of closed-cell StCu selected by visual inspection over the southeast Pacific Ocean: one group that transitions to POCs and another comparison group (CLOSED) that does not. The high spatio-temporal resolution of the new GOES-16 data allows for a detailed examination of the temporal evolution of POCs in this region. We find that POCs tend to develop near the coast, last tens of hours, are larger than $10^4$ km$^2$, and often (88%
of cases) do not re-close before they exit the StCu deck. Most POCs are observed to form at night and tend to exit the StCu during the day when the StCu is contracting in area. Relative to the CLOSED trajectories, POCs have systematically larger effective radii, lower cloud drop number concentrations, comparable conditional in-cloud liquid water path, and a higher frequency of more intense precipitation. Meanwhile, no systematic environmental differences other than boundary-layer height are observed between POC and CLOSED trajectories. Interestingly, there are no differences in reanalysis aerosol-optical depth
between both sets of trajectories which may lead one to the interpretation that differences in aerosol concentrations are not influencing POC development or resulting in a large number that re-close. However, this largely depends on the reanalysis treatment of aerosol-cloud interactions and the product used in this study has no explicit handling of these important processes. These results support the consensus view regarding the importance of precipitation on the formation and maintenance of POCs and demonstrate the utility of modern geostationary remote sensing data in evaluating POC lifecycle.

## 1 Introduction

Stratocumulus (StCu) often organize into two distinct regimes: closed and open cells. Closed cells tend to have a higher albedo (McCoy et al. 2017) and greater cloud fraction (CF; Rosenfeld et al., 2006) than open cells, while open cells tend to produce more intense precipitation, especially at the edge of open cells (Stevens et al., 2005; Wood et al., 2008, 2011; Eastman et al.,

2019; Sarkar et al., 2019). It is commonly observed that regions of open cells can develop within closed-cell Sc, and these
regions have been defined as pockets of open cells (POCs; Bretherton et al., 2004; Stevens et al., 2005). POCs are often
long-lived (10s of hours to a few days) once they have formed (e.g. Stevens et al., 2005; Berner et al., 2013).

A large body of evidence suggests that POCs are subject to a similar large-scale environment as closed-cell StCu that never
transition to open cells. For example, several studies have found that the inversion height within POCs is similar to the nearby
closed-cell region (Sharon et al., 2006; Bretherton et al., 2010; Berner et al., 2011, 2013). Sharon et al. (2006) used aircraft
observations from the Drizzle and Entrainment Cloud Study coupled with GOES-10 geostationary observations to analyze the
characteristics of rifts of open cells embedded within the StCu deck. They found that the boundary layer within both the rift and
surrounding cloud is well-mixed with similar moisture profiles above the boundary layer. Using a mixed-layer model and large-
eddy simulations (LES), Bretherton et al. (2010) hypothesized that POCs and the surrounding StCu have inversion heights that
are "symbiotically" locked together through dynamical coupling even though the two adjacent regions may experience different
local cloud-top entrainment rates. Studies have also noted that POCs tend to be advected by the mean flow (i.e., same direction
and at the same speed) along with the surrounding StCu (Stevens et al., 2005; Sharon et al., 2006) and in-situ data indicates that
wind shear within the POCs and the surrounding cloud are similar (Comstock et al., 2007). Taken together, the observations
and theory indicate that transitions of closed-cell StCu to POCs are not driven by differences in large-scale meteorological
forcing but are instead driven by processes internal to the boundary layer.

Precipitation is widely viewed as a key process in the development and maintenance of POCs. Prior observationally-based
studies have found that POCs form at night (Wood et al., 2011; Burleyson et al., 2013; Burleyson and Yuter, 2015) when the
most intense precipitation in the closed-cell StCu deck begins to cluster and organize (Comstock et al., 2007; Savic-Jovcic and
Stevens, 2008; Wang et al., 2010b; Glassmeier and Feingold, 2017). In response, cold pools develop that drive more intense
updrafts and precipitation (Wang and Feingold, 2009; Terai and Wood, 2013; Yamaguchi and Feingold, 2015; Ghate et al.,
2020), especially at the boundary between open and closed cells (Stevens et al., 2005; Wood et al., 2008), which act to both
reduce CF by depleting cloud water (Austin et al., 1995) and drive the entrainment of drier air at cloud top (Comstock et al.,
2005). Observational studies have also found the air inside POCs tends to have lower aerosol concentrations (Petters et al.,
2006; Szoeke et al., 2009; Wang and Feingold, 2009; Wood et al., 2011; Terai et al., 2014), which results in a more conducive
environment for precipitation by reducing the number of cloud droplets while increasing their size (Wood et al., 2011; Abel
et al., 2017; Watson-Parris et al., 2021). Feingold et al. (2015) used a cloud-resolving model to show that aerosol concentrations
in the cloud layer are especially important for the end of a POC's lifetime. They found that there needs to be an injection of
aerosols into the POC which can allow the generation of cloud water to overcome precipitation loss causing the POC to close,
which can be achieved through events like biomass burning (Abel et al., 2020; Gupta et al., 2021).

To perform a process-based analysis of POC transitions, a model is needed. However, observations can be used to infer
processes influencing POCs such as precipitation by analyzing the differences in cloud microphysical properties between
POC regions and the surrounding closed-cell clouds. Prior observational-based studies have primarily used field-campaign
observations to analyze POCs (e.g. Petters et al., 2006; Szoeke et al., 2009; Wood et al., 2011; Terai et al., 2014). Most
satellite-based studies have primarily focused on differences between closed- and open-cell StCu (e.g. Painemal and Zuidema,

2010; Goren et al., 2018; Eastman et al., 2021), but a few have focused on the characteristics of POCs and the surrounding Sc.

Of note, Wood et al. (2008) used a combination of shipborne remote sensing coupled with MODIS and GOES-8 observations to analyze changes in cloud drop number concentration and liquid water path within POCs, finding that POCs tend to form in regions of higher liquid water path and lower number concentrations. Recently, Watson-Parris et al. (2021) composited MODIS observations of POCs between 2005 and 2018 to analyze POC characteristics. They found POCs have a larger effective radius, lower cloud optical depth, and smaller cloud water path than the surrounding cloud.

A foundational aspect of Wood et al. (2008) was their use of GOES-8 to investigate POC development. By using geosynchronous observations, they were not limited to instantaneous snapshots of POCs from instruments such as MODIS (e.g. Eastman and Wood, 2016; Watson-Parris et al., 2021). Wood et al. (2008) found that two-thirds of the POC cases identified between September and October 2001 formed in the early morning hours when cloud drop and aerosol concentrations were lowest. However, the cloud microphysical characteristics were not derived from GOES-8; instead, Wood et al. (2008)

qualitatively compared the GOES-8 visible, near-infrared, and infrared observations to MODIS-derived retrievals and aircraft observations. Abel et al. (2020) made quantitative use of the Spinning Enhanced Visible and Infrared Imager (Aminou, 2002) onboard the Meteosat Second Generation geosynchronous satellites and MODIS to investigate the influence of biomass burning on POCs. They found that the boundary layer within POCs is ultra-clean even in columns containing aerosols emitted from biomass burning, suggesting that open-cellular convection does not efficiently entrain free-tropospheric aerosols from

immediately above the inversion into the boundary layer.

In this study, we add a Lagrangian perspective of the full POC lifecycle from satellites by using GOES-16 passive measurements of StCu in the southeast Pacific (SEPAC). GOES-16 makes full-disk observations at 10-minute time intervals with a horizontal resolution between 0.5-2 km. The continuous observations afforded by a geostationary orbit allow for a characterization of POC cloud properties throughout the POC lifetime. Furthermore, the data allow for a comparison of the cloud

properties along Lagrangian trajectories that develop into POCs with similar trajectories that remain closed-cell. We use these observations to demonstrate that POCs and closed-cell StCu experience indistinguishable large-scale forcing yet markedly different cloud microphysical properties, thereby supporting the consensus view regarding the role of precipitation in POC formation and maintenance.

## 2   Data and Methods

### 2.1   POC Identification

We identify POCs visually by creating true-color RGBs using 0.47-$\mu$m, 0.64-$\mu$m, and 0.86-$\mu$m reflectance during the day and 10.3-$\mu$m − 3.9-$\mu$m brightness temperature difference (TB$_{10.3\mu m - 3.9\mu m}$) images at night using Satpy (Raspaud et al., 2018) from 10-minute observations of GOES-16 ABI top-of-atmosphere solar reflectance and infrared brightness temperatures (IR; Schmit et al., 2017). The 0.64-$\mu$m reflectance is sampled at 0.5 km, while 0.47-$\mu$m and 0.86-$\mu$m reflectances are sampled at 1

90  km so we resample the 0.64$\mu$m reflectance to 1 km resolution before creating the true-color RGBs. The 3.9-$\mu$m and 10.3-$\mu$m TBs are sampled at 2 km. We focus on the southeastern Pacific Ocean (SEPAC) defined as the region spanning 45°S – 5°N

and 70°W – 120°W during September – November of 2019, and we create weekly animations of the true-color RGBs and $TB_{10.3\mu m \check{} 3.9\mu m}$ images to visually identify POCs. For consistency, we classify any region of clearing completely enclosed within the StCu deck as a POC with the following conditions: 1) regions of clearing at the StCu edge that become completely

enclosed within the StCu deck are not classified as POCs, and 2) any potential POCs that we visually identify to develop in response to gravity waves (Allen et al., 2013) are not included. POC development due to gravity waves are excluded because they close very quickly and move in the direction of wave propagation instead of with the mean wind. Visual identification is admittedly subjective, and the overall classification of POCs under this framework may slightly differ by person. However, our overall results, as discussed later, are consistent with prior POC studies and a subjective approach is common in this literature

(e.g. Wood et al., 2008, 2011; Terai et al., 2014; Watson-Parris et al., 2021).

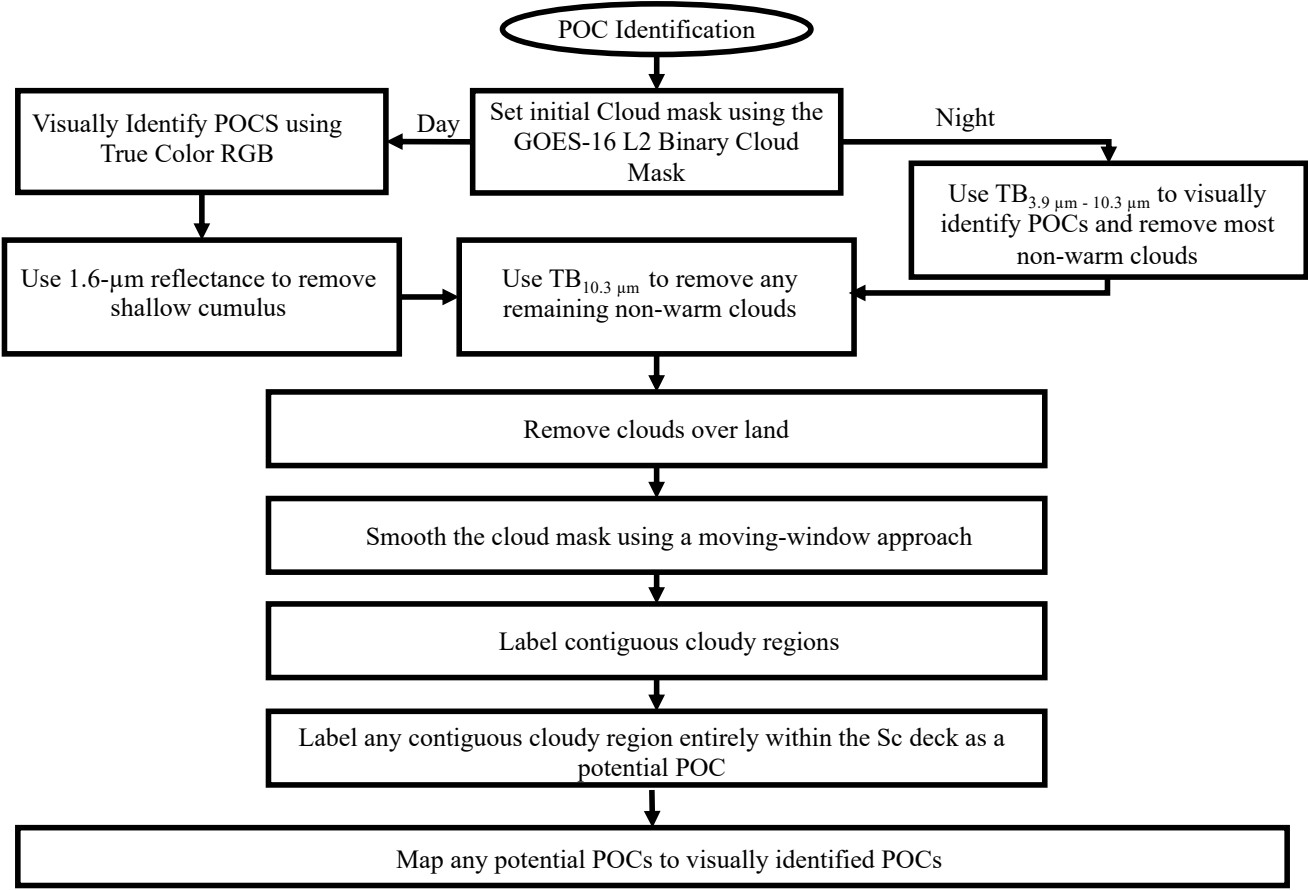

**Figure 1.** Flowchart of how POCs are identified.

Once we visually identify POCs, we develop an overcast-StCu mask that is used to track the evolution of each POC within the larger cloud field. As shown in Figure 1, the initial overcast-StCu mask is defined using the clear-sky mask level-2 product from GOES-16 by filtering out clear pixels (Heidinger and Straka, 2012). The next step is different depending on if there is daylight. During the day, 1.6-$\mu$m reflectance ($R_{1.6}$) (resampled from 1 km to 2 km) is used to filter out ice-phase clouds and many shallow cumulus clouds (Cu) from the overcast-Sc mask. $R_{1.6}$ has two useful tendencies in this regard. First, because water droplets are more reflective at 1.6-$\mu$m (Miller et al., 2014), $R_{1.6}$ tends to be larger for water droplets than ice crystals. Second, $R_{1.6}$ tends to be brighter in StCu than cumulus pixels (e.g. Zinner and Mayer, 2006; Wolters et al., 2010). To exploit these tendencies, we find the median $R_{1.6}$ of all cloudy pixels within the SEPAC region and exclude the lowest 50% of remaining cloud pixels that tend to be associated with ice-phase and liquid phase cumulus clouds (Figure 2c-d). At night, the initial filter uses $TB_{10.3\mu m \check{} 3.9\mu m}$ as an initial overcast-StCu mask filter. Considering that warm cloud emissivity is smaller at 3.9-$\mu$m than ice cloud emissivity but similar at 10.3-$\mu$m (Hunt, 1973), $TB_{10.3\mu m \check{} 3.9\mu m}$ has been used to separate both cloud types (e.g. Jedlovec et al., 2008). Therefore, any pixels with $TB_{10.3\mu m \check{} 3.9\mu m} < 0.3$ K are excluded (Figures 3c-d). Once the overcast-StCu mask is conditioned using either $R_{1.6}$ or $TB_{10.3\mu m \check{} 3.9\mu m}$, we remove any remaining cold clouds using a threshold 10.3-$\mu$m TB of 273 K and clouds over land using the Global 1-km Base Elevation dataset (Hastings and Dunbar, 1999).

The filters above effectively precondition the overcast-StCu mask but it still likely includes the brightest Cu and it can be noisy near the edge of the StCu deck. To smooth the overcast-StCu mask, we calculate the mean binary mask value within a 25x25 pixel window centered on each pixel. We filter out any overcast-StCu mask pixels with a window-mean mask value < 0.5 (Figures 2i-j and 3i-j). We chose this threshold because, upon visual inspection, it does not over-smooth the overcast-StCu mask by removing the edge of POCs that are close to the StCu edge but does effectively remove large and bright Cu.

From Figures 2i-j and 3i-j, we see that there are clear/variable cloudy regions within the overcast-Sc. Therefore, any contiguous clear/variable cloudy region completely enclosed within a contiguous overcast region is classified as a potential POC (Figures 2k and 3k). We then map the potential POCs to the observed cloud field to visually confirm if a potential POC is meeting our subjective POC criteria defined above. Comparing Figure 2a to 2l and Figure 3a to 3l, we see that our algorithm can effectively identify both StCu decks and the POCs they enclose. We identify a total of 258 distinct POCs that form/end between September and November 2019.

## 2.2 Defining Lagrangian Trajectories

We use trajectory analysis to track the evolution of each POC. We define the initial POC time by identifying a time when each POC is larger than 3136 km$^2$ (approximately one 0.5°x0.5° gridbox) and visually not too irregularly shaped such that the POC centroid is contained within the POC region and not in the surrounding StCu. From this time and location, we run forward and backward trajectories using the nearest 3-hourly 925-mb horizontal winds from Modern-Era Retrospective analysis for Research and Applications, Version 2 (MERRA-2; Gelaro et al., 2017), with a time step of 10 minutes to follow the Lagrangian evolution of POCs. The trajectories are run forward and backward from 10-minutes before POC development up to 6 hours before POC development and 6 hours after POC dissipation. If any trajectory intersects any other POC in the 6-hour timeframe before POC development or after POC dissipation, it is terminated prematurely because our goal is to understand the temporal

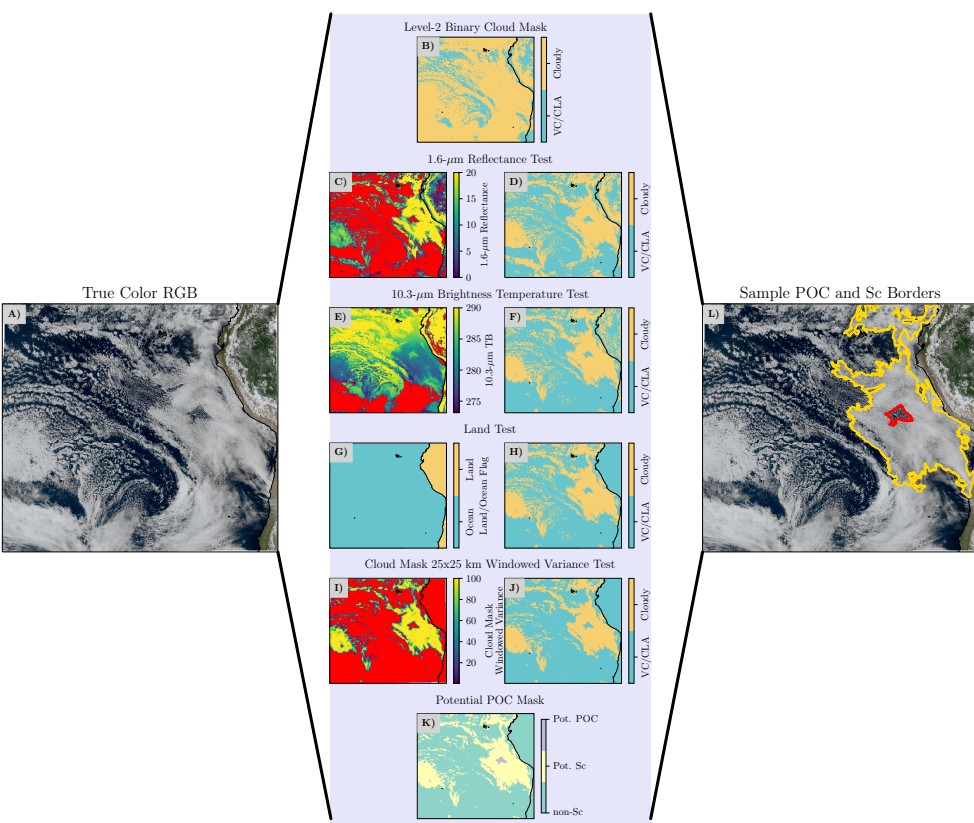

**Figure 2.** Shown above are the steps used to identify pockets of open-cell (POCs) stratocumulus during the day. In the third column, the stratocumulus deck borders are in yellow and POC borders are in red.

transition from overcast StCu to POCs. We do not expect this to substantially impact our results, because it represents only 7% of all pre-POC and 5% of all post-POC hours.

We classify the POC start time as the time along each trajectory when CF begins to decrease. POC end time is determined using two separate criteria: 1) CF increases back to 100% and does not change for more than an hour (hereby known as the POCs that re-close) or 2) at least one POC edge reaches the edge of the StCu deck (hereby known as the POCs that never re-close). The first criterion is based on the calculated changes in CF along each trajectory, while the second criterion must be satisfied visually. Throughout the rest of the paper, these trajectories will be labeled as POC cases/trajectories.

To compare closed-cell StCu that develop into POCs and closed-cell StCu that do not, we run additional trajectories initialized at the same starting location of each POC trajectory at $\pm24$-hour intervals until we find a comparison (CLOSED) trajectory that does not intersect any POCs to ensure both trajectories travel through similar meteorological regimes at the same time of day. Note, that this is only an approximate control because it is impossible to obtain a true control in which all variables that set CF are exactly equivalent. Using a 24-hour interval between POC and CLOSED trajectories result in 147

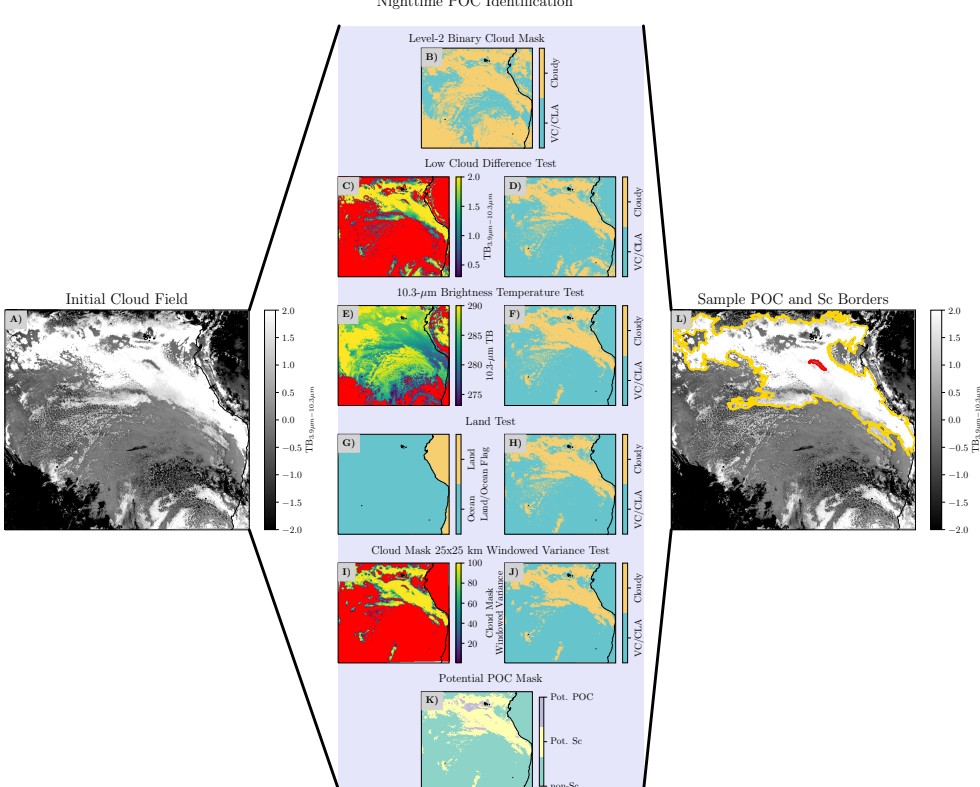

**Figure 3.** Shown above are the steps used to identify pockets of open-cell (POCs) stratocumulus during the night. In the third column, the stratocumulus deck borders are in yellow and POC borders are in red.

of the 258 POCs identified having a valid comparison. Similar to the POC trajectories, we define a before, during, and after time for the comparison trajectories. Specifically, we define the "before" time as the time prior to the comparison trajectory reaching the location where the POC forms, and we define the "after" time as the time after the comparison trajectory reaches the location where the POC dissipates (for the group of POCs that re-close, open cells transition back to closed cells, and for the group that never re-closes, open cells start transitioning to Cu).

## 2.3 Cloud Properties

We compare the following cloud properties along the POC and CLOSED trajectories: CF, cloud optical depth (COD), cloud top effective radius (re), liquid water path (LWP), and cloud drop number concentration (N). COD, $r_e$, LWP, and N are composited from cloudy pixels within a $0.5°$ x $0.5°$ window surrounding each trajectory point. This window size is close to the same size as a MERRA-2 gridbox, and our results are not sensitive to window size. Finally, CF is defined as the number of cloudy pixels divided by the total number of pixels within each window. As mentioned prior, different channels and algorithms are used to retrieve COD and $r_e$ during the day and night. During the day, a combination of 0.64-$\mu$m and 2.25-$\mu$m reflectance are

used (Walther et al., 2013), while 3.9-$\mu$m, 11.2-$\mu$m, and 12.3-$\mu$m brightness temperatures are used at night (Minnis and Heck, 2012). The day/night retrieval algorithms are fundamentally different. At night, COD is limited from 0 to 16 and $r_e$ is limited to 2 $\mu$m – 78 $\mu$m, whereas, during the day, COD can be retrieved from 0.25 to 158 and $r_e$ can be retrieved from 2 $\mu$m to 100 $\mu$m. The dynamic range is smaller at night because the emissivity of larger particles is similar at 11.2-$\mu$m and 12.3-$\mu$m, resulting in a smaller range of COD and $r_e$ values that can be discerned (Lin and Coakley, 1993). An effect of the limited range of nighttime optical depth retrievals is that a noticeable fraction of nighttime CODs are exactly 16. The diminished range of COD and $r_e$ limits our analysis of cloud-property differences between the POC and CLOSED trajectories. Therefore, we focus only on the daytime cloud property results in the main text and use the nighttime results as a comparison dataset as discussed in Appendix A. The GOES-R algorithm switches between day and night at a solar zenith angle of 82°, but any retrievals during twilight (solar zenith angles from 65° – 90°) are degraded (Minnis and Heck, 2012; Walther et al., 2013) and are therefore removed from the analysis. LWP is derived using equation 8 from Wood (2006), and N is derived using equation 9 from Wood (2006).

Prior studies have also found biases in retrieved optical properties in broken clouds due to cloud-edge effects (Coakley et al., 2005; Vant-Hull et al., 2007; Platnick et al., 2017; Zhu et al., 2018) and 3D radiative transfer artifacts (Zhang et al., 2012; Liang et al., 2015). GOES-R does not flag partially-filled pixels similar to MODIS (e.g. Jensen et al., 2008), meaning these biases likely influence our results. Therefore, we only include microphysical properties (COD, $r_e$, LWP, and N) on cloudy pixels connected to four other cloudy pixels (excluding corners), similar to MODIS (Platnick et al., 2017), to limit these biases.

Another potential limitation of the data is cirrus contamination. To account for this, we use different sets of tests that are not used in the POC identification algorithm, during the day and night to remove cirrus. During the day, cirrus removal is based on the GEOS 1.37-$\mu$m channel (Schmit et al., 2018), and all cloud-property values within any 0.5°x0.5° window containing any 1.37-$\mu$m reflectance values < 5 are removed. We subjectively chose this value, because we visually found a 1.37-$\mu$m reflectance threshold of 5 results in the lowest amount of non-cirrus cloud removal, while removing the most cirrus. We also use 8.4-$\mu$m - 10.3-$\mu$m brightness temperature difference ($TB_{8.4\mu m \smile 10.3\mu m}$) to remove cirrus because $TB_{8.4\mu m \smile 10.3\mu m}$ tends to have positive values for ice clouds and small negative values for low water clouds (Baum et al., 2000; Giannakos and Feidas, 2013); previous studies (Krebs et al., 2007; Strandgren et al., 2017) have removed retrievals using similar thresholds of $TB_{8.4\mu m \smile 10.3\mu m}$. This algorithm effectively removes thick cirrus; however, it struggles to remove thin cirrus. Despite this, we find any potential influence of thin cirrus does not affect the overall statistics discussed in our results. Note, these tests are not used to flag cirrus clouds moving over the StCu deck that our POC identification algorithm may identify as a potential POC, because we can visually distinguish cirrus from POCs in the weekly animations.

## 2.4 Precipitation

We compare precipitation intensity along both the POC and CLOSED trajectories using the Advanced Microwave Scanning Radiometer 2 (AMSR-2) precipitation product constructed using the routine developed in Eastman et al. (2019). This dataset is based on statistical relationships between 4 x 6 km$^2$ AMSR-E 89 GHz microwave brightness temperatures and collocated CloudSat rain rates, applied to 3 x 5 km$^2$ AMSR-2 observations. These statistics are used to derive the probability of precipitation and area-averaged-CloudSat-like precipitation across the microwave radiometer swath, which allows for significantly

more potential overlap with GOES-16 identified POC cases than does CloudSat. We co-locate AMSR-2 with GOES-16 by identifying any time within 20 minutes of a GOES-16 timestamp where AMSR-2 precipitation is observed within any 0.5° x 0.5° POC box. Note, because AMSR-2 is polar orbiting and only available at most twice a day, these co-locations between AMSR-2 and GOES-16 only provide limited snapshots of precipitation along each trajectory, typically at times near the equator crossing time of the A-Train satellite constellation, at 0130 and 1330 local time. Figure 4a visually demonstrates this, showing variations in matched precipitation within an example POC. However, due to the statistical nature of the AMSR-2 product, we find that the data identify 98% of all POC and CLOSED trajectory AMSR-2 pixels as possibly raining over SEPAC, which is much higher than the typical rain fractions we found of 5% using the rain certain classification from CloudSat. To correct this, we use a precipitation threshold of 0.1 mm day$^{-1}$, which corresponds to the probability of rain reaching the surface typically below 5% (Figure 4b) and is consistent with the minimum observable value of surface rain (Comstock et al., 2004; Rapp et al., 2013).

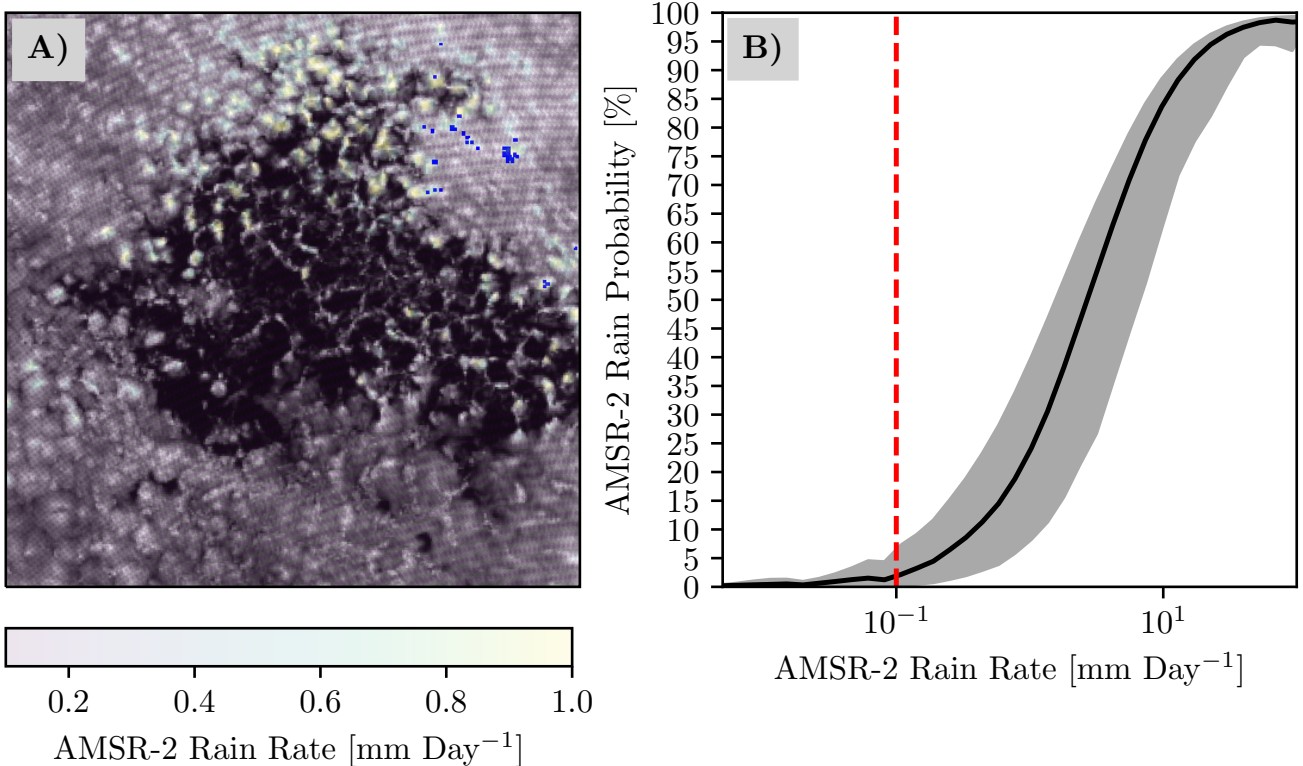

**Figure 4.** Advanced Microwave Scanning Radiometer 2 (AMSR-2) rain rates matched to the POC shown in Figure 2 overlaid on GOES-16 0.64-$\mu$m reflectance are shown in panel A). AMSR-2 rain probability as a function of rain rate for all September-November 2019 data over the entire domain are shown in panel B), where the solid black line represents the median probability at a given rain rate, grey fill represents the $10^{th}$-$90^{th}$ percentile spread at a given rain rate, and the dashed red line represents the rain rate threshold of 0.1 mm day$^{-1}$.

## 2.5 Environmental Conditions

We classify the large-scale environment at each point along the POC and CLOSED trajectories using sea-level pressure (SLP), estimated-inversion strength (EIS), 700-mb water vapor mixing ratio ($q_v$), 700-mb omega ($\omega$), planetary-boundary layer (PBL) height, PBL mean $q_v$, lifted condensation level (LCL) height, aerosol-optical depth (AOD), 50-m winds, and 925-mb wind speed/direction, all of which are derived from the nearest MERRA-2 grid point. EIS is calculated using equation 4 from Wood and Bretherton (2006), where lower-tropospheric stability (LTS; Slingo, 1987; Klein and Hartmann, 1993) represents the difference between potential temperature at 700 mb and 2-m temperature is taken from MERRA-2. The 850-mb moist adiabatic lapse rate ($\Gamma_m^{850}$) is derived using Metpy (May and Bruick, 2019). The LCL height is derived using MERRA-2 2-m temperature and the formulation of Romps (2017). MERRA-2 outputs two PBL Height variables, one based on the total-eddy diffusion coefficient of heat (PBLH), and another based on the bulk Richardson number (TCZPBL; McGrath-Spangler and Molod, 2014). Ding et al. (2021) found both generate PBL depths shallower than those derived directly from satellites, but PBLH is closer. Therefore, we use PBLH as our proxy for PBL height.

## 3 Results

### 3.1 General POC Characteristics

Of the 147 POCs that have valid comparison trajectories, Figure 5a shows that most POCs traverse between 5°S-25°S, and 80°W-100°W, similar to prior satellite-based studies (Wood et al., 2008; Watson-Parris et al., 2021). In comparison, Figure 5b shows the CLOSED trajectories take similar paths to the POC trajectories, increasing our confidence that both the POC and CLOSED trajectories experience similar meteorology during their lifetimes.

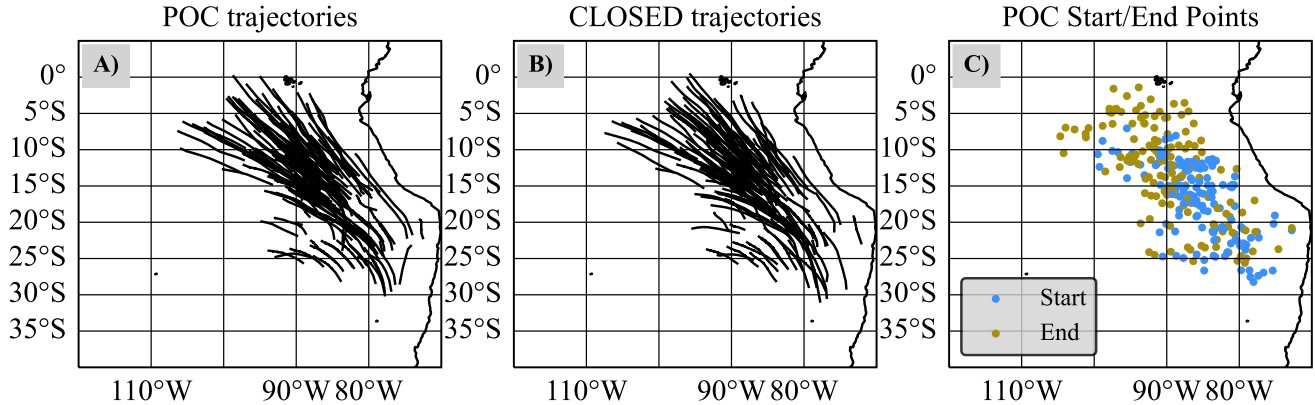

**Figure 5.** Modern-Era Retrospective analysis for Research and Applications, Version 2 trajectories that traverse closed-cell stratocumulus and develop into pockets of open cells are shown in panel A), the MERRA trajectories that traverse through closed-cell stratocumulus and never develop into pockets of open cells are shown in panel B), and POC starting and ending points are shown in Panels C).

Interestingly, 88% of all POCs never re-close, meaning that they remain open-celled until they exit typically north/northwest of the StCu deck. Further breaking this down, 129 POCs never re-close, 12 POCs re-close, and 6 of the calculated trajectories leave their associated POC area prematurely. Note that the 6 POC trajectories that prematurely exit POCs are not included in the remainder of this analysis. Here we note that we could use the POC centroids themselves to define the trajectory to salvage these discarded trajectories. However, that method would only work during the POC lifetime whereas the use of reanalysis trajectories allows us to extend the trajectories both before and after the POC lifetime. Using a cloud-resolving model, Feingold et al. (2015) found that the recovery from open- to closed-cell StCu is much slower than the transition from closed- to open-cell StCu, and it depends on the replenishment of aerosols resulting in cloud water increases exceeding precipitation loss. Our results suggest this does not frequently happen in the SEPAC, and as a result, POCs that never re-close and those that do are grouped together throughout the remainder of this paper.

Figure 5c shows the location where POCs typically begin and end, showing that POCs tend to form between 10°S-20°S, and 80°W-90°W and that POC starting locations tend to cluster more than POC ending location. This implies that there is relatively high variability in POC duration. To quantify this, Figure 6 shows a histogram of POC duration. POCs are generally long-lived and last an average of 20 hours, similar to previous observational (Stevens et al., 2005) and modeling (Berner et al., 2013) results.

Table 1 shows that most POCs form at night and end during the day, and have a median duration typically between 10 and 30 hours. Further demonstrating this, Figure 7 shows the diurnal cycle of the relative frequency of POC start and end times, with POCs typically forming at night between 0 - 6 local time and ending during the day between 9 and 13 local time. This is consistent with findings by Wood et al. (2008) that showed POC formation peaks around 3 local time when precipitation is most intense and StCu are thickest (Wood et al., 2008; Burleyson et al., 2013). Why might POCs preferentially end (exit the StCu) during the day? We find that StCu area reaches a minimum around 12 local time (Figure S1). As a result, we hypothesize that the tendency for POCs to exit the StCu during the day may simply be the result of a general reduction in StCu extent during the daylit hours (Burleyson and Yuter, 2015), so that the StCu edge effectively moves towards the POC during sunlight. This hypothesized mechanism is related to the tendency for the marine boundary layer to decouple from the surface during the daytime (e.g. Burleyson et al., 2013; Wilbanks et al., 2015) which also reduces the likelihood of POCs re-closing during the day.

## 3.2 POC Growth

Figure 8a shows the relative frequency distribution of median POC area, with values typically reaching $4.6 \times 10^3$-$4.7 \times 10^4$ km$^2$. Figure 8b shows changes in POC median area as a function of POC duration. POC area does not depend on when they start or end (Table 1) but tend to grow larger the longer they last, with POCs lasting $> 30$ hours having an average area approximately 3.6 times larger than POCs lasting $< 10$ hours. We could conclude that longer-lived POCs have more time to grow larger. However, 76% of distinct POCs that form end up merging at least once, which also contributes to increase POC area over time. Figure 8c shows the relative frequency of average POC growth rate after visually removing mergers, which cause artificial spikes in the growth rate. The median POC growth rate is 133 km$^2$ Hour$^{-1}$ with interquartile range between 42 and 422

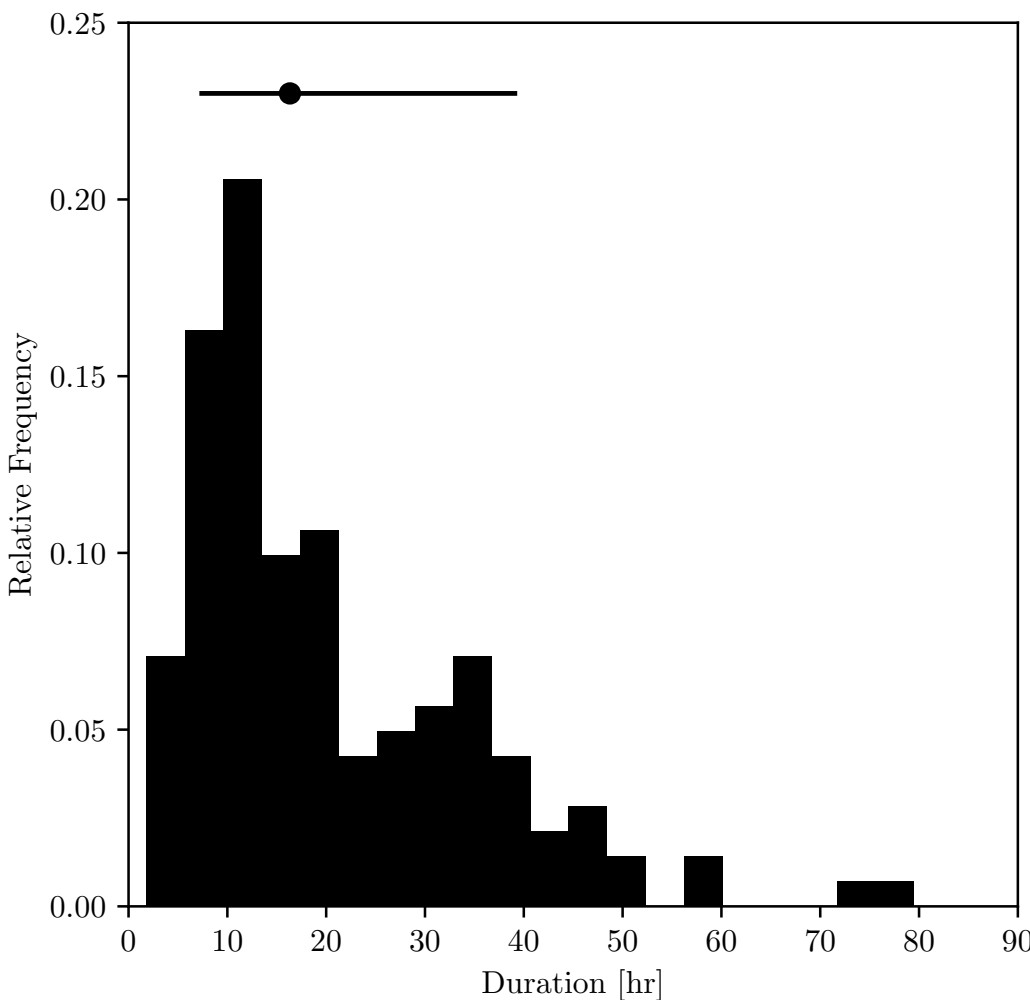

**Figure 6.** A histogram of the duration of pockets of open-cumlus stratocumlus is shown. The black dot above the histogram represents the median and the horizontal black line represents the $10^{th}$ to $90^{th}$ percentile spread.

km$^2$ Hour$^{-1}$. Figure 8d shows that the shortest-lived POCs occasionally grow faster than the longest-lived POCs, however the medians are not notably different. Furthermore, the results for smaller POCs may be biased due to limited sampling after removing mergers, problems during twilight, and cirrus contamination. We conclude that POCs tend to grow at a fairly consistent rate throughout their lifetime with significant departures from those rates associated with POC mergers.

### 3.3 POC Occurrence and Associated Environment

In this section the large-scale environmental conditions are contrasted between periods when POCs occur frequently and when they do not occur. For this analysis we include all 258 identified POCs. Figure 9 shows that there are usually between 0 and 7

**Table 1.** Shows total number of cases, median lifetime, and median area of pockets of open-cell stratocumulus (POCs) that start/end during the day, night or twilight (defined as solar-zenith angles between 65° and 90°) for valid POC cases (those that follow a POC along its entire trajectory).

|  | Number of Cases | Median Lifetime [hr] | Median Area [km$^2$] |
|---|---|---|---|
| Form at night and end during day | 77 | 12.5 | 15640 |
| Form at night and end at night | 9 | 30.7 | 18460 |
| Form at night and end at twilight | 8 | 8.9 | 8120 |
| Form at day and end during night | 6 | 18.3 | 66462 |
| Form at day and end during day | 27 | 22.3 | 15528 |
| Form at day and end during twilight | 3 | 39.3 | 17908 |
| Form at twilight and end during day | 8 | 26.8 | 17852 |
| Form at twilight and end during night | 2 | 11.0 | 56100 |
| Form at twilight and end during twilight | 1 | 36.2 | 17332 |

| Overall Statistics: | | | |
|---|---|---|---|
|  | Number of Cases | Median Lifetime [hr] | Median Area [km$^2$] |
| All Valid Cases | 141 | 16.3 | 18060 |

POCs on any given day. However, this activity is not random, with extended periods of frequent POC occurrence, followed by sustained periods with few POCs. Therefore, we created two separate groups: days when no POCs developed and days when > 7 POCs developed then composited the average environments for both groups as shown in Figure 10.

Setting the stage synoptically, we find that a surface high pressure system (Figures 10a and 10b) is present south of the geographic maximum CFs in both cases. On average, SLP is moderately lower during periods when the most POCs develop (Figure 10c), resulting in weaker southeasterly winds (Figure 10f) in the region of highest StCu CF. In both cases, there is a strong inversion with EIS typically > 5 K (Figures 10p and 10q) and the presence of lower-free-tropospheric subsidence (Figures 10s and 10t) which are conducive to StCu clouds. Interestingly, EIS is stronger over the southern part of the SEPAC but weaker where most POCs begin (Figure 10r).

On days with the most POCs, the PBL is on average shallower (Figure 10i), moister (Figure 10l), and with a lower LCL (Figure 10o). The lowering of PBL and LCL height have similar patterns and magnitudes suggesting a relatively constant cloud layer depth between POC and non-POC days. Figure 10x shows AOD is generally lower, except very close to the continent, on days with the most POCs. This is consistent with prior studies that found POC air tends to be cleaner than non-POC air (e.g. Wood et al., 2011; Terai et al., 2014). However, the pattern of the composite difference is still complicated and we must

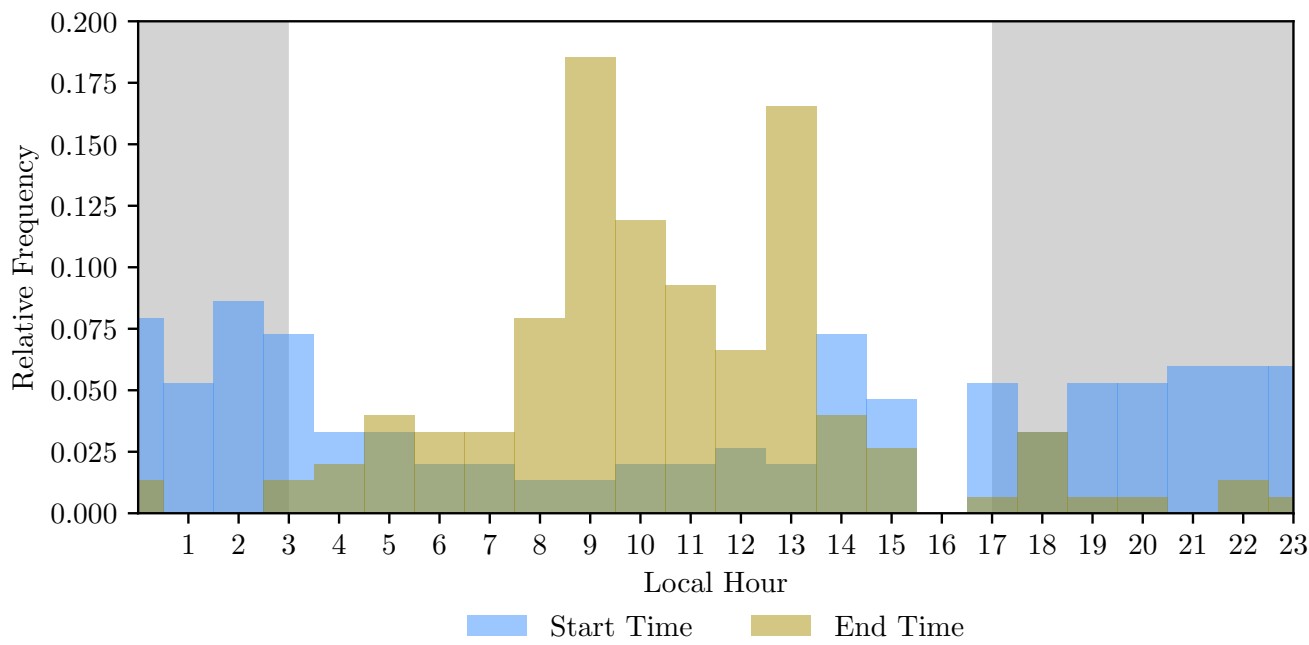

**Figure 7.** Histograms of the start time (blue) and end time (brown) of pockets of open-cell stratocumulus are shown.

exercise some caution in the interpretation of an aerosol field derived from a reanalysis. Furthermore, the differences between AOD, LCL height, and qv may be a symptom of a higher number of POCs, because POCs are cleaner, have lower LCLs, and more moisture near the surface compared to non-POC boundary layers.

Figure 11a shows that StCu area is typically larger when there are more POCs. On days with the most POCs, this implies that a shallower and moister boundary layer, lower LCLs, large-scale subsidence, and a stronger inversion promotes higher CFs (Wood, 2012) and larger StCu area (Figure 11b). As a result, the simplest explanation for why there are several days with no POCs and several days with many POCs may be that environmental conditions favoring StCu development result in a higher likelihood of POC development.

It is reasonable to be concerned about the robustness of these results, because of the limited MERRA-2 sample size during days with no and many POCs. To quantify the physical significance of the differences in mean environmental properties, we calculated a signal-to-noise ratio as $\frac{\mu_{many} - \mu_{none}}{\frac{1}{2}(\sigma_{many} + \sigma_{none})}$ where $\mu_{many}$ and $\sigma_{many}$ represent the mean and standard deviation of each distribution on days with $> 7$ POCs and $\mu_{none}$ and $\sigma_{none}$ represent the same statistics on days with no POCs. We then calculate a regional mean of this quantity, weighted by average low-CF, for each environmental characteristic. This value

ranges from 0.14-0.42 and is largest for the characteristics with the largest mean differences between days with no and many POCs (PBL height, $q_v$, and LCL height), suggesting that the mean differences in these characteristics are the most physically meaningful. However, we note that there is in general a great deal of overlap between the distributions of all of the environmental characteristics on the days with large numbers of POCs and the no-POC days. Overall, it is unclear what environmental

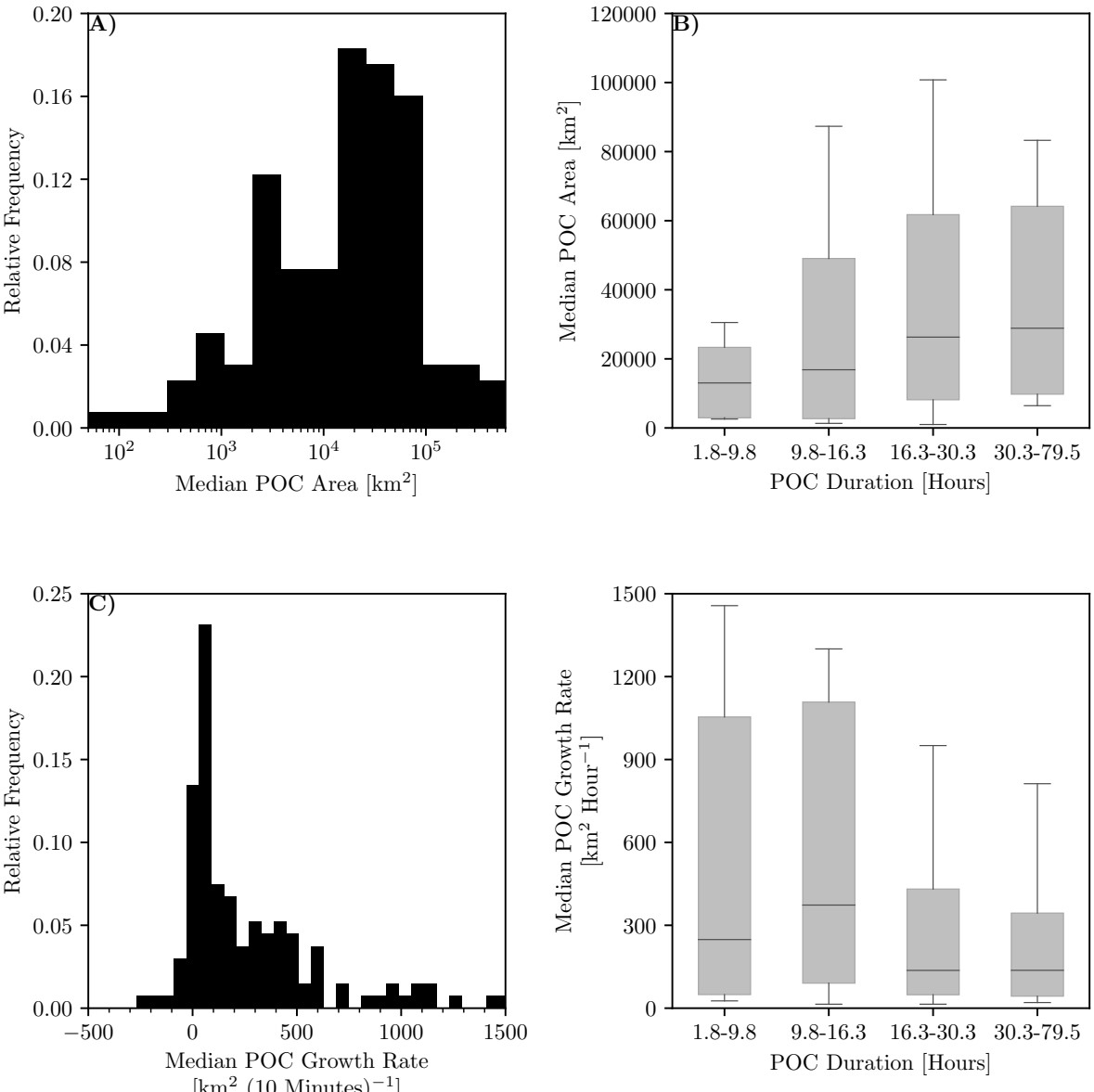

**Figure 8.** The Histogram of the median area of pockets of open-cell stratocumulus (POCs) is shown in panel A), the distribution of median POC area as a function of POC duration is shown as box plots in panel B), where the solid line represents the median duration, the Histogram of POC median growth rate is shown in panel C), and the distribution of median POC growth rate as a function of POC duration is shown as box plots in panel B). Note that in the boxplots, the shaded boxes represent the interquartile range, the solid lines inside the shaded regions represent the median, and the whiskers represent data between the $10^{th}$ and $90^{th}$ percentiles.

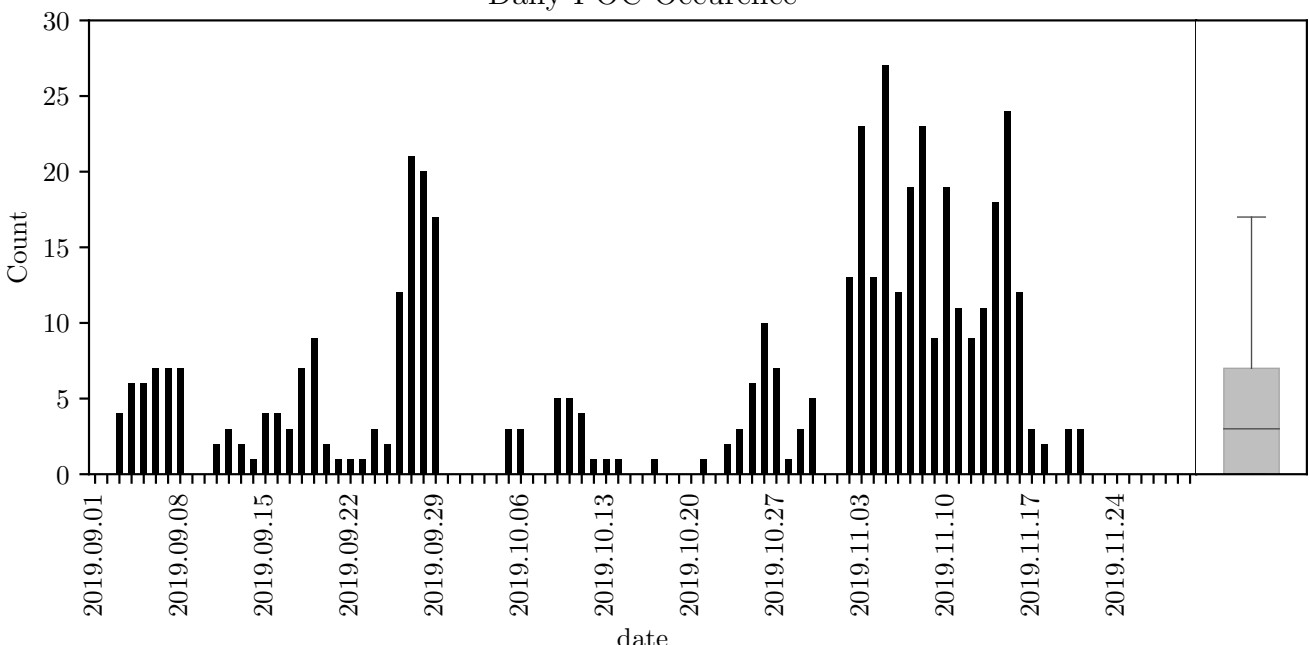

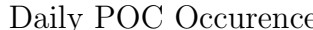

**Figure 9.** The number of pockets of open-cell stratocumulus (POCs) occurring on each day is shown. The boxplot represents the overall distribution of daily POC occurrence, where the solid line represents the median counts, the shaded box represents the interquartile range, and the whiskers represent data between the $10^{th}$ and $90^{th}$ percentiles.

conditions, outside of those favoring StCu development, favor POC development due to limited sample size of days with few
and many POCs. This is an area for a future investigation, which will require substantial increases in sample size.

### 3.4 Daytime Cloud-Field Characteristics

We now turn to a comparison of CF and cloud microphysical properties for the POC and comparison trajectories. Figure 12 shows the evolution of an example POC and CLOSED trajectories in terms of CF, average COD, average in-cloud LWP, average $r_e$, and average N. The largest changes along the POC trajectory are in CF, COD, $r_e$, and N relative to the CLOSED trajectory,
while LWP remains relatively constant. Specifically, Figure 12a shows that CF decreases as the POC develops. This decrease in CF is accompanied by decreasing COD (Figure 12b), increasing $r_e$ (Figure 12d), and decreasing N (Figure 12e). These variables then remain relatively constant during the POC's lifetime before approaching pre-POC values after POC dissipation. Larger $r_e$ and lower N inside of this non-re-closing POC are consistent with the development of precipitation, resulting in the closed- to open-cell transition (Comstock et al., 2007; Savic-Jovcic and Stevens, 2008; Wang et al., 2010a; Glassmeier and
Feingold, 2017).

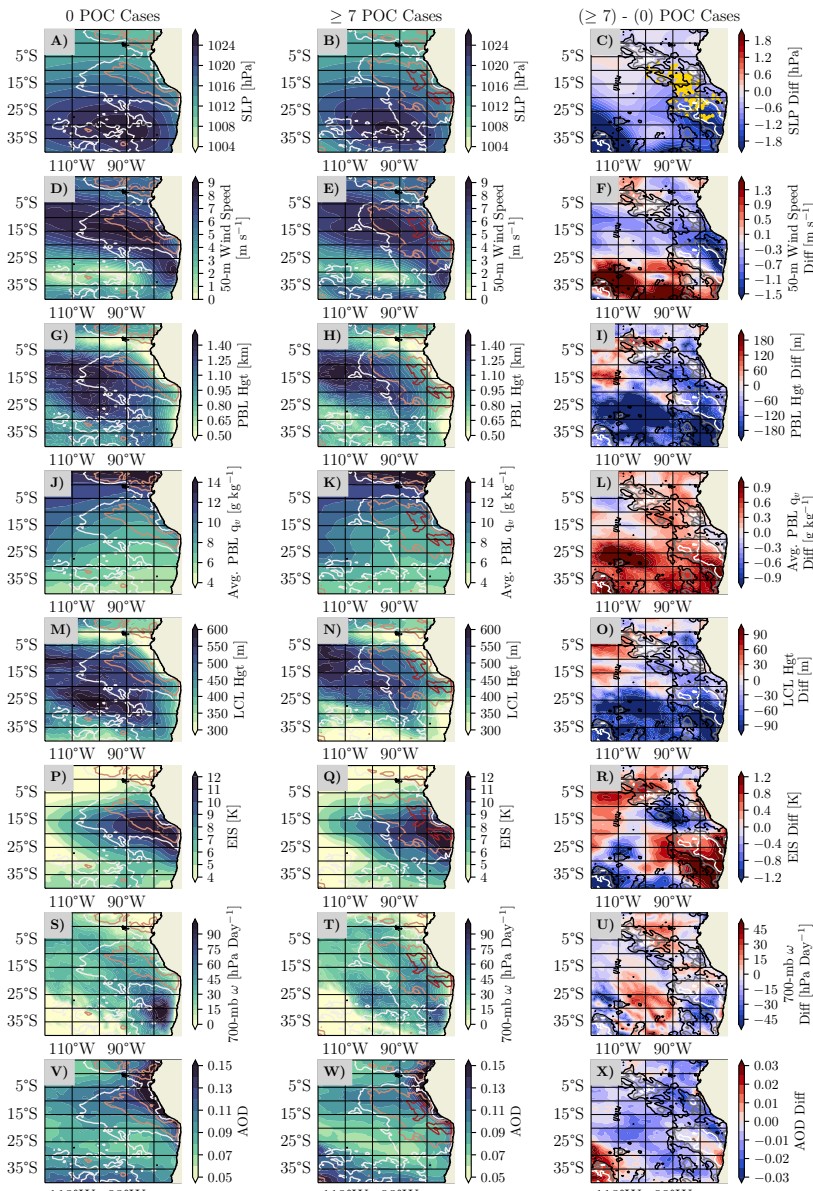

**Figure 10.** Average sea-level pressure (SLP; panel A), 50-m winds (panel D), planetary-boundary layer (PBL) height (panel G), average PBL water vapor mixing ratio ($q_v$; panel J), lifted-condensation level (LCL; panel M), estimated-inversion strength (EIS; panel P), 700-mb omega ($\omega$; panel S), and total-column aerosol-optical depth (AOD; panel V) are shown in the first column for days when no POCs occur. The second column shows the same variables for days when > 7 POCs occur. The third panel shows the difference between each variable for days with > 7 POCs and days with no POCs. The non-filled contours overlaid in all panels represent cloud fraction, where white represents 70%, orange represents 80%, and red represents 90%. The gold dots shown in panel C) overlay POC starting point.

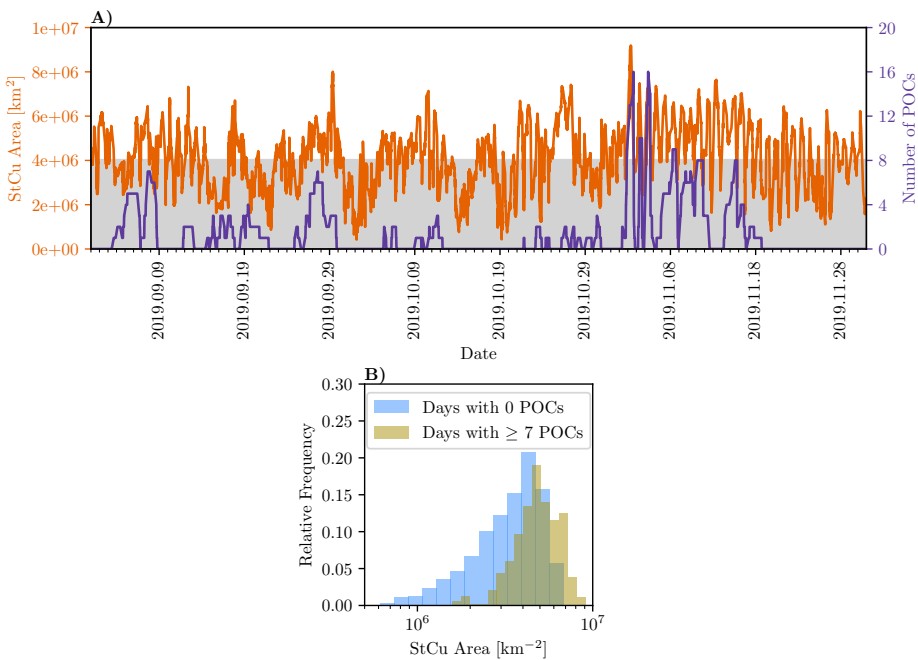

**Figure 11.** Changes in stratocumulus (StCu determined using 0.5-2-km level 1.5 observations) and number of pockets of open-cell StCu (POCs) every 10 minutes are shown in Panel A), and the distributions of area for days with no POCs (blue) and days with > 7 POCs (brown) are shown in Panel B).

One case gives us a glimpse at how cloud field properties vary throughout a POC's lifetime, and how they compare to a CLOSED trajectory. However, more cases are needed to make more definitive conclusions. For a more in-depth analysis, Figure 13a-e shows the trends of the cloud-field properties for the 6 hours before POC formation until 10 hours after formation for all POCs and the equivalent time-snippet for the CLOSED trajectories. Note that this is only daytime data due to retrieval differences during day and night (Section 2.3). The results shown are similar to Figure 12, with CF decreasing (Figure 13a), COD (Figure 13b), LWP decreasing (Figure 13c), $r_e$ increasing (Figure 13d), and N (Figure 13e) decreasing along the POC trajectories relative to the CLOSED trajectories. Increasing $r_e$ and decreasing N are consistent with an increased likelihood of precipitation (Stevens et al., 2005; Wood et al., 2008, 2011; Sarkar et al., 2019; Eastman et al., 2021). The overall differences in COD and LWP are not that large between the POC and CLOSED trajectories, however the slight decrease in LWP may be an indication of a loss of LWP due to precipitation flux in POCs. Interestingly, there is a bump in COD and LWP around 5 hours after POC development for the CLOSED trajectories. We suspect that this feature may result from the changing sample number shown in Figure 13f where the lowest number of samples (possibly due to cirrus contamination or time of day) occur just prior to this.

Further breaking this down into the overall statistics for the before, during, and after period, Figure 14 shows the distributions of the cloud field and precipitation characteristics of all POCs and their associated CLOSED trajectories represented by

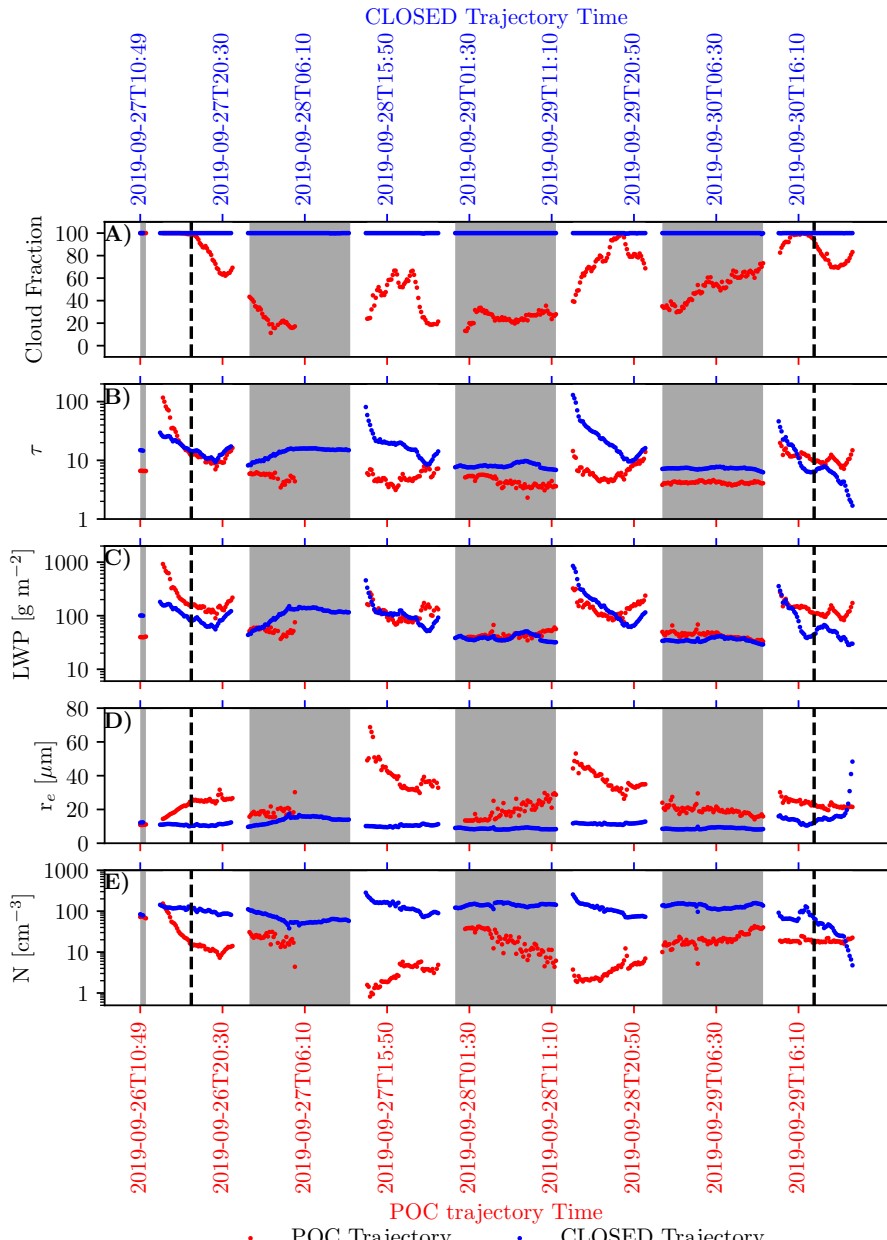

**Figure 12.** Changes in cloud fraction (panel A), cloud optical depth (panel B), liquid water path (panel C), effective radius (panel D), and cloud drop number concentration (panel E) are shown along the trajectory for one pocket of open-cell stratocumulus (POC) that occurred between 09/26/2019 and 09/29/2019. The red dots correspond to a sample POC trajectory, while the blue dots correspond to the corresponding CLOSED trajectory. Gray shading indicates night time periods.

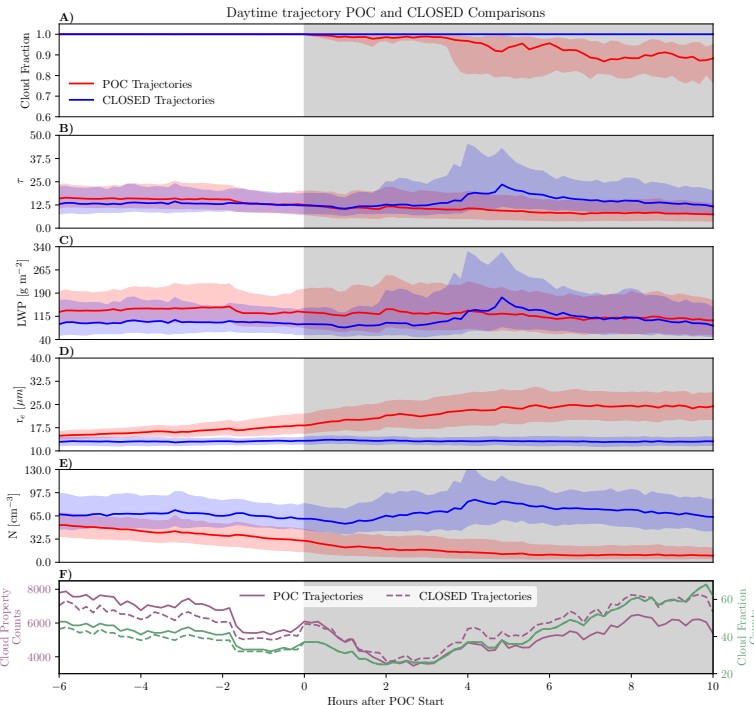

**Figure 13.** The median changes in daytime cloud fraction (panel A), cloud optical depth (panel B), liquid water path (panel C), effective radius (panel D), and cloud drop number concentration (panel E) for the six hours before until ten hours after POC development are shown along all pocket of open-cell stratocumulus (POC shown in red) and comparison (CLOSED shown in blue) trajectories, where the red and blue fill represent changes in the interquartile range. The number of valid samples for both the GOES-16 microphysical properties (purple) and GOES-16 cloud fraction (green) are shown in panel F). Gray shading indicates the time after POC development.

the median and $10^{th}$-$90^{th}$ percentile spread (see Figure S2 for the full daytime distributions). It confirms the differences in CF, $r_e$, and N are largest between the POC and CLOSED trajectories, with the most overlap in COD and LWP. The overall differences between the POC and CLOSED trajectories do not change much in the transition from the during-to-after periods, likely because most POCs never re-close before transitioning from a StCu to Cu cloud regime. The patterns in these different

properties are consistent with prior observational works (e.g. Wood et al., 2008, 2011; Terai et al., 2014; Watson-Parris et al., 2021), increasing our confidence that these differences are not retrieval artifacts due to broken cloud (Coakley et al., 2005). Although the magnitudes of the nighttime results are smaller (Appendix A), we found the overall patterns in each cloud property is similar increasing our confidence in these results. Finally, Table 2 shows the number of samples in the before, during, and after periods are sufficient to make robust conclusions, even though our study is limited to 3 months and the before and after

periods are limited to at most 6 hours.

**Table 2.** The number of daytime samples for GOES-16 cloud fraction, GOES-16 microphysics (cloud-optical depth, liquid-water path, cloud-top effective radius, and cloud-drop number concentration), and Advanced Microwave Scanning Radiometer 2 rain rates is shown.

| POC | | | |
|---|---|---|---|
| | Before | During | After |
| Cloud Fraction | 1475 | 5883 | 3381 |
| Microphysics | 243019 | 769595 | 392349 |
| Rain Rates | 4551 | 11810 | 8270 |
| Environment | 278 | 1161 | 263 |
| CLOSED | | | |
| Cloud Fraction | 1435 | 5749 | 3330 |
| Microphysics | 232877 | 937092 | 507102 |
| Rain Rates | 3271 | 8378 | 6698 |
| Environment | 278 | 1161 | 263 |

## 3.5 Precipitation Characteristics

Figures 14p-r show the precipitation statistics in a manner similar to the cloud statistics. To further illustrate the differences between POC and CLOSED precipitation, Figure 15 shows the distribution of rate-weighted rain rate. Specifically, each bin is multiplied by bin-center rain rate, which places a higher weight on bins with more intense precipitation such that the area under the histogram is equal to the accumulated rainfall.

Precipitation rates are typically low with median values between 0.14 and 0.44 mm day$^{-1}$ and they remain relatively constant from before POC development until after POC dissipation. Precipitation is generally more intense at night than during the day (Figure 14p-r) which is consistent with prior works (Wood et al., 2008; Burleyson et al., 2013). Notably, there is a larger spread in rain rate. Some of this spread may result in a much more limited number of AMSR-2 observations compared to GOES-16 (Tables 2 and S1), but our results showing more intense precipitation for the POC compared to the CLOSED trajectories is consistent with expectations. Although evident during both day and night, the differences in the occurrence of the most intense precipitation between the POC and CLOSED trajectories are most distinct before POC development at night. The precipitation is usually light for both the POC and CLOSED trajectories (Figure 14p-r), and Figure 15 clearly highlights that there is substantially more precipitation accumulation for the POC then the CLOSED trajectories during the POC lifetime. Together these finding are consistent with larger r$_e$ (Figures 14j-l) and lower N (Figures 14m-o) retrieved from GOES-16 for POCs than the CLOSED trajectories, which suggest that intensity and quantity of precipitation is key to the formation and maintenance of POCs.

## 3.6 Environmental Characteristics

In section 3.3, we evaluated the synoptic environment during periods of frequent POC occurrence and rare occurrence in an attempt to identify conditions conducive to POC formation. Most notably we found that the largest numbers of POCs tended

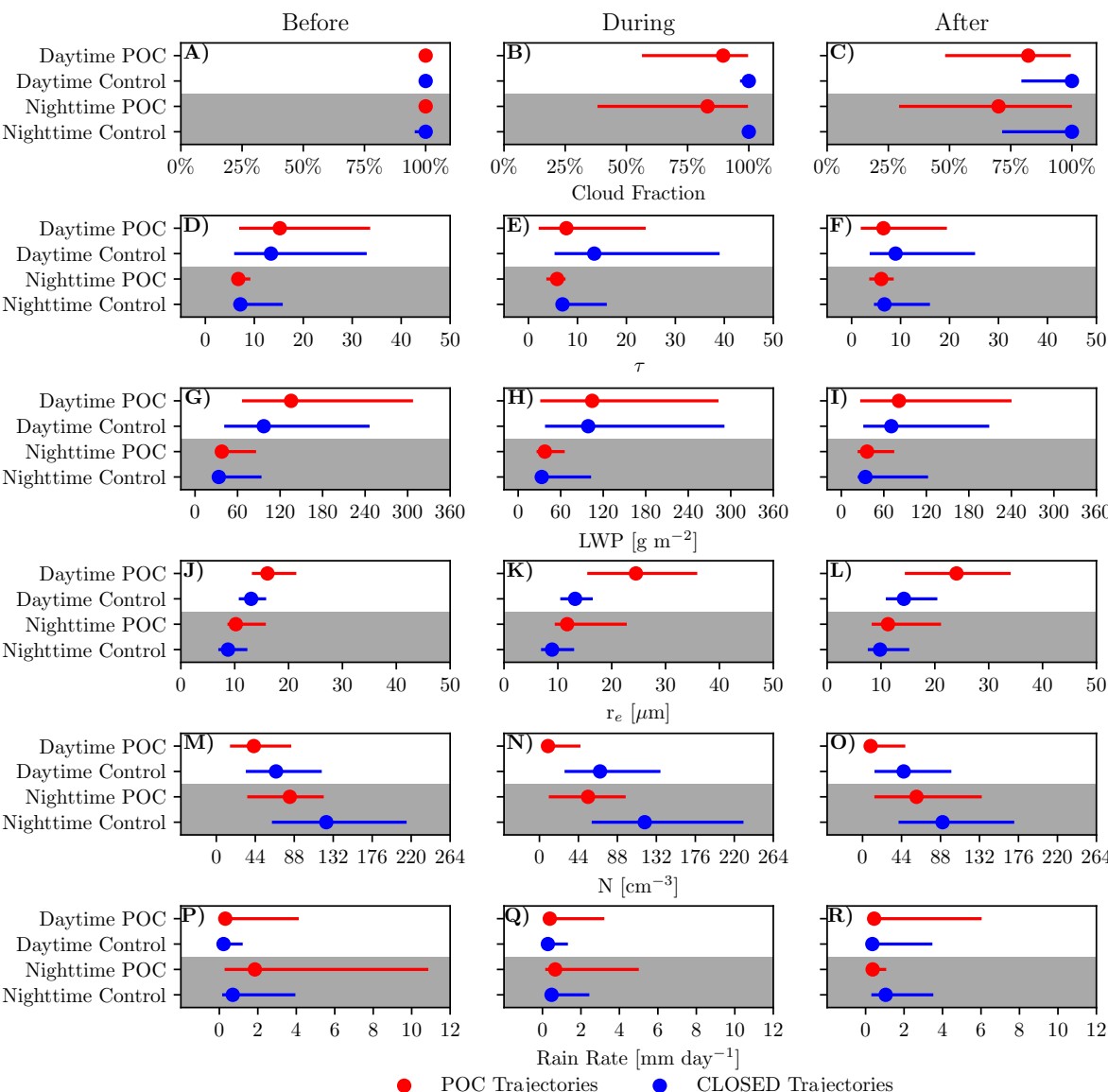

**Figure 14.** Daytime statistics for cloud fraction (A, B, and C), cloud-optical depth (D, E, and f), liquid-water path (G, H, and I), cloud-top effective radius (J, K, and L), cloud-drop number concentration (M, N, and O), and Advanced Microwave Scanning Radiometer 2 rain rate (P, Q, and R) are shown in the upper-white half of each panel, while the nighttime statistics are shown in the grey-lower half of each panel. Red values represent the pocket of open-cell stratocumulus (POC) trajectories, while the blue values represent the comparison (CLOSED) trajectories. The colored dots represent the median of each distribution, and the horizontal lines represent the spread between the $10^{th}$ and $90^{th}$ percentiles.

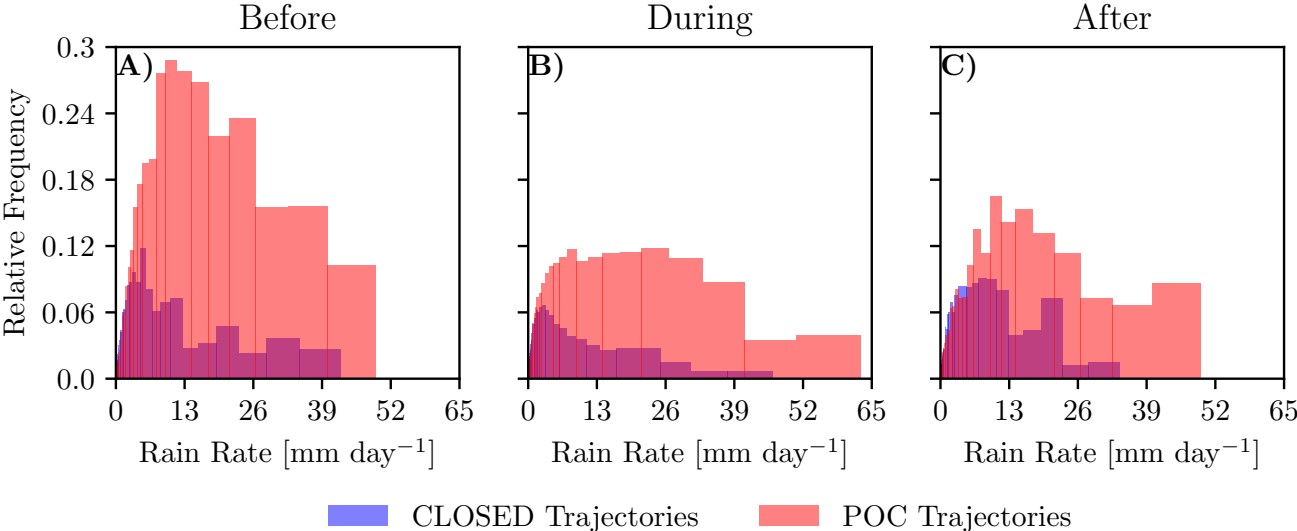

**Figure 15.** Histograms of Advanced Microwave Scanning Radiometer 2 rain rate along the pocket of open-cell stratocumulus (POC shown in red) and comparison (CLOSED shwon in blue) trajectories before (A), during (b), and after (c) POC lifetime are shown, where each bin is multiplied by bin-center rain rate (e.g. accumulation weighted) and the area under each curve is normalized to mean rain rate for each distribution.

to form when AOD is low and the StCu area is high. Here we will instead evaluate the differences in the environmental conditions along the POC with the CLOSED trajectories to identify whether there are systematic differences between the large-scale forcing experienced by the two trajectories. We focus primarily on PBL characteristics because of their potential importance of rain to POC development. We find that PBL height (Figures 16a-c), mean moisture (Figures 16d-f), LCL height (Figures 16g-i), AOD (16j-l), and 925-mb winds (Figure 17) are similar for both the POC and CLOSED trajectories, with PBL height around 800-1400 m, moisture around 7-11 g kg-1, AOD around 0.05-0.15, LCL height around 400-600 m, and 925-mb winds from the southeast at  20 knots. Results from Sharon et al. (2006) are similar to ours showing that PBL depth and 925-mb winds are similar in both open- and closed-cell environments, however they found the open-cell PBL is moister. This is likely because changes in cloud microphysics and PBL turbulence interact with the environment changing the moisture structure in the cloud layer, and the aircraft measurements Sharon et al. (2006) used likely captured these differences. The MERRA-2 PBL characteristics we show are likely more related to the large-scale environment, because small-scale processes are not well constrained in MERRA-2 (e.g. Witte et al., 2021). Additionally, we find there are no significant differences between the POC and CLOSED trajectories in EIS, 700-mb $q_v$, and 700-mb $\omega$ (Figure S4). Throughout a POC's lifecycle, similar environments are consistent with observational results showing POC occurrence does not depend on free-tropospheric moisture (Wood et al., 2008), and modeling (Berner et al., 2011)and observational (Sharon et al., 2006; Wood et al., 2011) results found

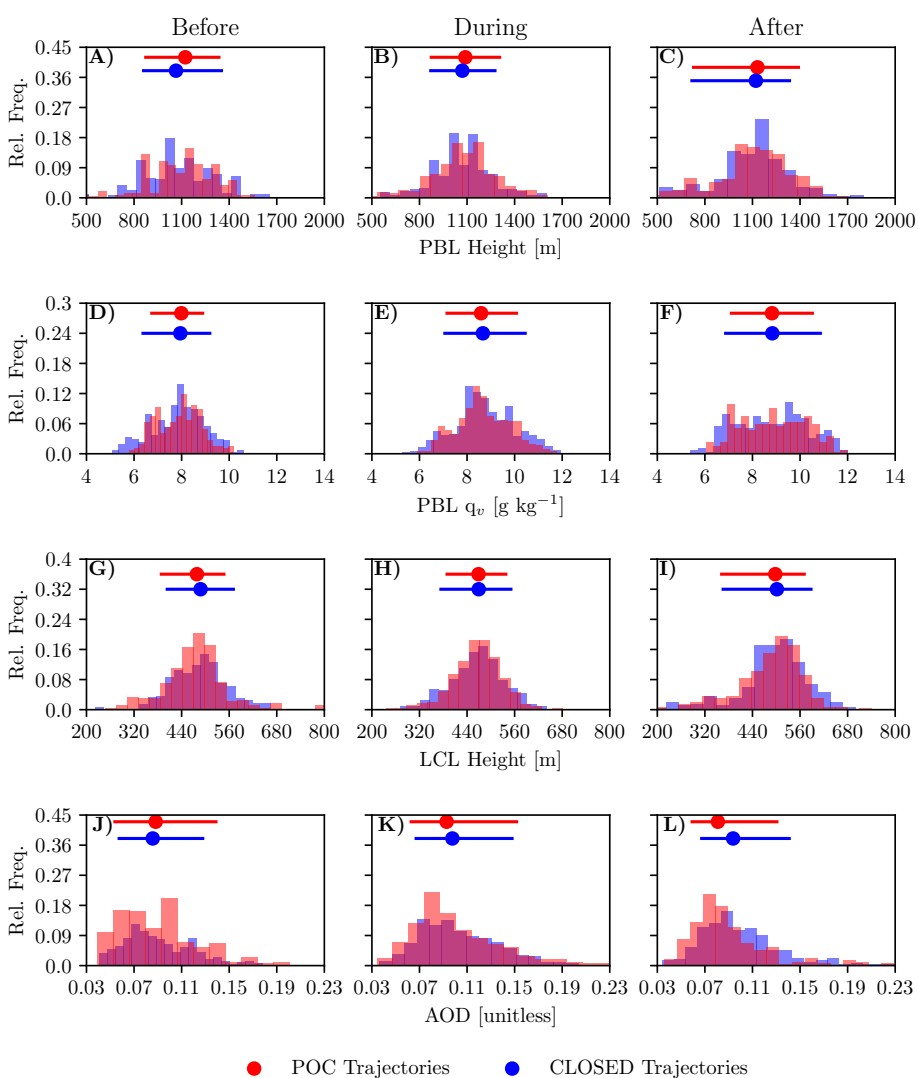

**Figure 16.** Distributions of environmental properties before a POC forms (left column), during POC lifetime (middle), and after POC dissipation (right). Top row (A, B, and C) shows planetary boundary layer (PBL) height. PBL-average water vapor mixing ratio $q_v$ is shown in the second row (D,E, and F). Lifted-condensation level height is shown in the third row (G, H, and I). Aerosol optical depth is shown in the fourth row (J, K, and L). Red represents distributions mapped to the POC trajectories, while blue represents the distributions mapped to the CLOSED trajectories. The colored dots represent the median of each distribution, and the horizontal lines represent spread between the $10^{th}$ and $90^{th}$ percentiles.

little difference in moisture and potential temperature profiles between POCs and surrounding StCu. Prior studies have noted that off-shore flow from SEPAC may suppress open-cell development by introducing more aerosols into the environment (e.g. Wood et al., 2008; Abel et al., 2020), but our results indicate no aerosol difference between both sets of trajectories (Figure

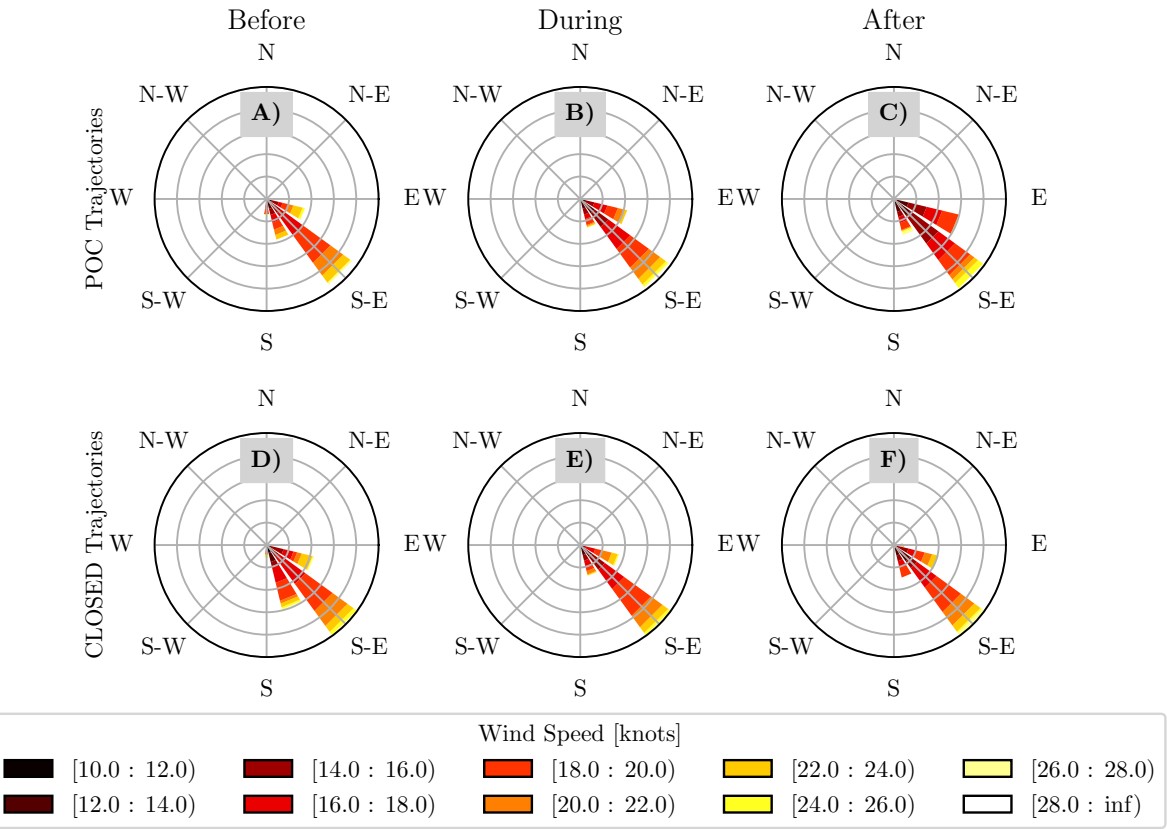

**Figure 17.** Wind rose plots are used to show the most common 925-mb wind speed and direction before a POC (panels A and D), during a POC (panels B and E), and after a POC ends (panels C and F). The wind rose plots in the top row represent the trajectories that intersect a POC, while those in the bottom row represent the trajectories that do not intersect a POC.

16j-l) suggesting our trajectories originated over ocean. Additionally, no aerosol difference between both sets of trajectories suggests the processes (i.e. aerosol injection leading to POC closure) hypothesized by Feingold et al. (2015) are not occurring among this set of POCs, however this is highly dependent on the treatment of cloud-aerosol interactions in MERRA-2. Although the number of MERRA-2 samples relative to the other datasets are low (Table S2), these results are consistent with the expectation that few differences exist between large-scale forcing where POCs form and those where they do not. This implies that the process relevant to the formation and maintenance of POCs are small-scale processes within the PBL.

## 4  Summary and Discussion

This study develops a novel methodology to identify POCs and then uses a Lagrangian analysis to track their evolution and how that equates to comparison trajectories of closed-cell StCu that never transition, and expands upon prior polar-orbiting-studies

that were unable to analyze the temporal evolution of POC characteristics. Along the trajectories, we analyze the cloud field characteristics, and environmental characteristics of both cases. We find that POCs tend to last on average 20 hours, have a maximum area larger than $10^4$ km$^2$, and exit the StCu deck without re-closing 88% of the time. POCs have a median growth rate of 133 km$^2$ Hour$^{-1}$ with an interquartile range of 42 – 422 km$^2$ Hour$^{-1}$. POC mergers are common, which result in sudden increases in POC area well above these typical growth rates.

We find that POC development and maintenance are most highly correlated with processes associated with the cloud microphysical state such as precipitation (Wood et al., 2011; Burleyson et al., 2013; Burleyson and Yuter, 2015; Eastman et al., 2021) as opposed to large-scale forcing, which does not appear to have an impact on POC formation (Sharon et al., 2006; Bretherton et al., 2010; Berner et al., 2011, 2013). Modeling studies (Feingold et al., 2010; Yamaguchi and Feingold, 2015) have shown and observational studies (Terai and Wood, 2013; Ghate et al., 2020) have inferred that POC formation may be related to the clustering of raining closed-cell StCu that drives the development of interacting cold pools, which subsequently causes more intense precipitation and initiates the transition to open cells. Even though we cannot reliably observe changes in precipitation organization alone from GOES-16, our results show that r$_e$ is typically higher, N is lower, more precipitation falls and is more intense for POCs than the CLOSED population before POC formation. These findings are consistent with precipitation occurrence and intensity driving likely cold-pool development and subsequent reductions in CF. Yamaguchi and Feingold (2015) used a cloud-resolving model to show that the distance between precipitating StCu cells is important to open-cell development. After POC formation our results show that more frequent rain persists, while r$_e$ becomes even larger and N decreases further than before POC formation. It is, however, also possible that at least some of the observed increases in r$_e$ during the POC period could be related to retrieval artifacts associated with increasing sub-pixel heterogeneity (Zhang et al., 2012; Liang et al., 2015).

Overall, these results appear robust and suggest precipitation as a key driver of POC development. However, we found evidence, similar to Allen et al. (2013), suggesting gravity waves may also influence POC development. We found this happened infrequently over SEPAC during September-November, 2019. Therefore, more observations over a longer timeframe are needed to assess their influence.

The general understanding of broader StCu to cumulus transitions is that they occur when StCu drift over warmer sea surfaces, leading to deeper and decoupled boundary layers (Albrecht et al., 1995; Wyant et al., 1997; Bretherton and Wyant, 1997; Stevens, 2000; Wood and Bretherton, 2004), in which cumulus clouds begin to develop that penetrate the overlying StCu layer and mix drier free-tropospheric air into the cloud layer (e.g. Wyant et al., 1997). These studies have argued that precipitation is not necessary for these transitions to occur, however there has been some debate over the importance of precipitation in the timing of the transition (e.g. Paluch and Lenschow, 1991; Yamaguchi and Feingold, 2015; Eastman and Wood, 2016; de Roode et al., 2016; Yamaguchi et al., 2017). Considering most POCs identified in this study never re-close, our results suggest that the development of POCs driven by organizing precipitation can mediate the timing of the stratocumulus to cumulus transition. Yamaguchi et al. (2017) used LES to investigate the influence of precipitation on StCu transitions, finding that, when aerosol/drop number concentrations are low due to precipitation, the StCu transition can be rapidly accelerated. Furthermore, as the open-cells within POCs continue to organize and precipitation intensity increases, the StCu deck transitions to a precipitating

Cu field (Yamaguchi et al., 2017). These results suggest a process possibly explaining why most POC cases we observe never re-close, but more observational studies tracking POCs over a longer timeframe and across more regions are needed to draw more general conclusions.

The results herein have important implications for aerosol-cloud-interactions and climate change mitigation through marine cloud brightening. Importantly, current models do a poor job of representing the warm rain process (e.g. Sun et al., 2006; Kharin et al., 2007; Wehner et al., 2014; Christopoulos and Schneider, 2021; Witte et al., 2021) and the scavenging effect of precipitation on aerosol (e.g. Tost et al., 2010; Grandey et al., 2014; Gettelman et al., 2015; Michibata et al., 2019; Jing et al., 2019). Our finding that POCs catalyze the StCu to Cu transition and prevents POCs from re-closing which suggests that the ability of aerosol to enhance cloud albedo is highly dependent on the current state of the cloud field. Specifically, the efficiency of cloud condensation nuclei at influencing cloud albedo over the cloud lifetime would be maximized prior to the nascent formation of drizzle that precedes POC development.

Even though our analysis is limited to a 3-month period over the SEPAC and may not generalize to other marine stratocumulus-dominated regions of the globe (i.e., northeast Pacific and southeast Atlantic basins). This study demonstrates, most importantly, that the improved spatio-temporal resolution of the current generation of geostationary sensors and the associated data product suite provides an important tool in evaluating the temporal dimension of POCs for future studies.

*Code availability.* Please contact the authors for access to any dataset created by the analysis and/or the code used to process the GOES-16/AMSR-2 data..

*Data availability.* The GOES-16 level-1 and level-2 data are available at https://registry.opendata.aws/noaa-goes/.

**Appendix A:  Cloud-Field Characteristics**

Despite the limited dynamic range of nighttime cloud-property retrievalss, they provide an independent comparison to the daytime cloud property results through the use of different channels (3.9-$\mu$m, 11.2-$\mu$m, and 12.3-$\mu$m brightness temperatures). Cloud properties retrieved at night (solar zenith angles > 90°) with $TB_{10.3\mu m\check{}3.9\mu m} < 0$ K (cirrus removal; Jedlovec et al., 2008) are analyzed and compared to the daytime results here.

Figures A1 and 14 (see Figure S3 for the full nighttime distributions) show that the overall patterns in CF, COD, $r_e$, N, and how they compare to the CLOSED trajectories are similar to the daytime characteristics of all POCs, with one main difference. The COD and derived LWP are substantially smaller than in the daytime data, and the CLOSED and POC LWP are essentially indistinguishable at night. This is likely an artifact of the limited dynamic range of the GOES-16 nighttime algorithms, resulting in inaccurate magnitudes. Nevertheless, given the substantial algorithmic differences in retrieved COD and $r_e$ at night versus during the day, it is encouraging that the microphysical patterns (i.e. $r_e$ and N) are similar. In particular, one might be concerned

that the daytime differences between the POC and CLOSED trajectories are merely an artifact of 3D radiative transfer artifacts differentially affecting the two regimes (Zhang et al., 2012; Liang et al., 2015). However, the fact that these signals are also observed in the emission-based nighttime data lends credence to their sign, while uncertainty in their magnitude remains.

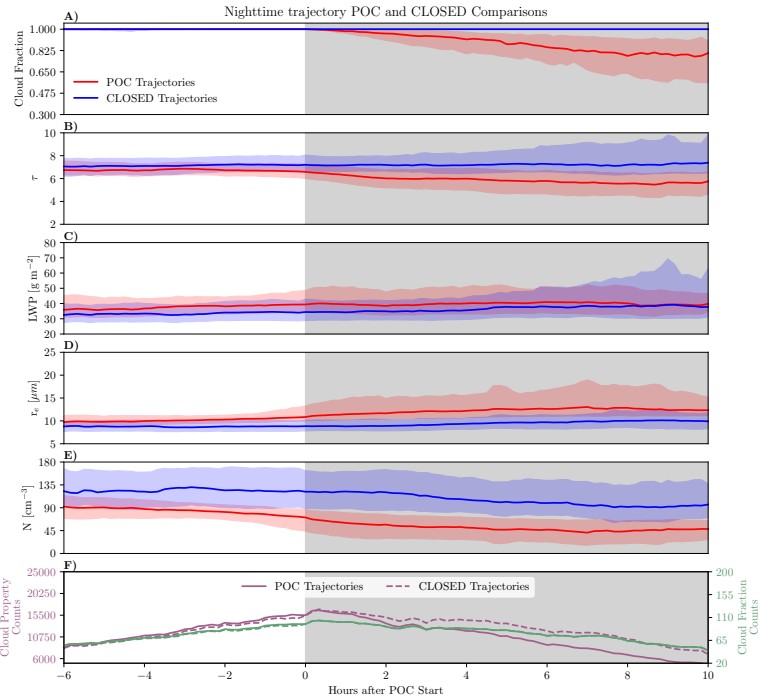

**Figure A1.** The median changes in nighttime cloud fraction (panel A), cloud optical depth (panel B), liquid water path (panel C), effective radius (panel D), and cloud drop number concentration (panel E) for the six hours before until ten hours after POC development are shown along all pocket of open-cell stratocumulus (POC shown in red) and comparison (CLOSED shown in blue) trajectories, where the red and blue fill represent changes in the interquartile range. The number of valid samples for both the GOES-16 microphysical properties (purple) and GOES-16 cloud fraction (green) are shown in panel F). Gray shading indicates the time after POC development.

*Author contributions.* KS performed the analysis presented in this paper. RE provided the AMSR-2 precipitation data. KS prepared the paper with contributions from ML, RE, MS, and MW.

*Competing interests.* The authors declare that they have no conflict of interest.

*Acknowledgements.* This work was performed at the Jet Propulsion Laboratory, California Institute of Technology, under a contract with the National Aeronautics and Space Administration and was funded by the CloudSat mission. We thank Dr. Rob Wood for several helpful discussions. MKW was supported in part by the US Department of Energy's Atmospheric System Research, an Office of Science Biological and Environmental Research program, under Grant No. DE-SC0020332. RE is supported by NASA grant 80NSSC19K1274.

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
