# Peer review of "A Lagrangian Analysis of Pockets of Open Cells over the Southeast Pacific"

_Atmospheric Chemistry and Physics, 2021_

## Referee Comment (RC2)

Review of Smalley et al "A Lagrangian Analysis of Pockets of Open Cells over the Southeast Pacific"

The manuscript provides an observational analysis of pockets of open cells (POCs) over the southeast Pacific during a three-month period using retrievals from the geostationary satellite GOES16 and other satellite/reanalysis products. Once the POCs are identified, the authors use MERRA2 reanalysis winds to track how their size and cloud and environmental properties change during the POC evolution, from before its formation to after its demise. The authors also run trajectories that start 24 hours before or after initial POC formation in cloudy overcast scenes to compare how the POC trajectories differ from a "control" trajectory case where no POCs form.

Although there have been a couple previously published, satellite-based analysis of POCs (Wood et al. 2008; Watson-Parris et al., 2021), the novel aspect of this study is the use of Lagrangian trajectories to examine the temporal aspect of clouds, particularly how they form and how they dissipate. Information of their temporal duration and area are new and interesting. However, I found that much of the manuscript focuses on the results that have been confirmed by past observations (more rain rate, larger cloud drops, lower aerosol or cloud droplet number concentration in POCs) rather than highlighting the new findings and its implications. I believe that the authors can make significant contributions to answering what conditions favor POC formation and what sets the size of POCs, which other observational studies have not been able to discuss at length. In the current manuscript, the discussion of these results seems minimal. In addition, there are a number of errors and issues associated with the methods and points of clarification that I would like to see the authors address.

Given that some of the methodological fixes potentially impact results and interpretations, I recommend major revisions before the manuscript is published.

Major comments

Lack of emphasis and discussion on the temporal evolution of POCs
The authors point out the novelty of their analysis is in capturing the temporal evolution of POC formation, maintenance, and dissipation. And based on the title, I expected to see more emphasis and discussion on such themes. However, most of the results of the paper seemed to focus on confirming past findings about the POC vs non-POC cloud and environmental conditions that have already been reported by previous studies, which did not use a Lagrangian perspective. That POCs have less clouds, larger cloud droplets, less aerosol/cloud drops, more intense rain rates, and that they have similar large-scale meteorological contexts as non-POC regions have been reported previously by examining POC conditions and the regions surrounding POCs, without looking at a Lagrangian perspective.
I believe that the authors are however well positioned and should focus on answering some questions that other studies have only lightly touched on or have not addressed. First, they can better answer the origins of POCs and whether there are any environmental precursors that make it much more likely for POCs to form. For example, are there necessary conditions for POC formation? The authors touch on this in Figure 13 and 14, and mention that there appear to be no environmental precursors to POC formation, but do not elaborate further about what it means

with respect to what previous studies have found (e.g., Wood et al., 2008; Yamaguchi and Feingold, 2015) and whether the authors find evidence to support or go against what has been proposed in those studies.

The authors are also well positioned to answer the downsides of relying on polar orbiting satellites to identify and characterize POCs. Do we miss anything by just observing POCs twice a day, except for the lack of temporally tracking their evolution?

The other set of questions that the authors are able to address relates to the growth of POCs. For example, what sets the growth of POC size? And for larger POCs, do they grow rapidly or steadily over time? Are POC sizes just determined by how long it takes for the POCs from formation to reach the edge of the stratocumulus deck? When POCs grow, do they have a preferred direction of growth relative to the 925hPa trajectory?

LTS & LCL calculations

For the LCL and LTS calculations, the surface air temperature is typically used, rather than the surface temperature (Romps, 2017; Klein and Hartmann, 1993). For the LTS, it seems that the sea surface temperature is used and potentially also for the LCL. I suggest the authors redo the analysis for these. Based on previous studies that have shown surface air temperatures tend to be colder in POCs due to precipitation evaporation, I would expect a lower LCL and stronger LTS over POCs vs non-POCs.

Furthermore, my impression from the reading the manuscript is that the boundary layer qv and LCL are frequently discussed as environmental factors (alongside large-scale subsidence or sea level pressure differences), when internal cloud and boundary layer processes (precipitation and boundary layer decoupling) can have significant impact on these variables. Please provide further explanation for why the authors consider them as environmental factors mainly determined by the large-scale meteorology, rather than internal boundary layer processes.

Analyses in Section 3.2

The synoptic comparison between no-POC and frequent (>7) POC days is interesting, but how robust are the results? In other words, how similar are the synoptics among frequent POC days vs no POC days compared to the difference between frequent and no POC days? There is no discussion about how large the POC vs non-POC differences are compared to the variance within each category.

It was also not clear to me what specific question the authors want to answer by examining the synoptic and environmental conditions. My impression is that the purpose of this section is to ask whether there are certain environmental conditions that favor POCs vs just common stratocumulus clouds. The statement at the end of the section (L250-254), which points out that many of the synoptic differences are consistent with conditions that favor more stratocumulus seems to muddy whether those synoptic differences we observe in the difference plots are indeed conducive for POCs specifically. Can the authors perform more in depth analysis with the available data (or by including other months) to solely compare non-POC days when there are large StCu cloud decks to see whether there are certain conditions that actually favor POC formation?

Note also that some things like, lower AOD, lower LCL, and higher qv might be a symptom of, rather than a cause behind having many POCs during a day, because POCs are cleaner, have lower LCL, and moisture near surface values compared to non-POC boundary layers.

Specific/minor questions

L8 – Although it also shows up in the conclusions, I believe that 104km2 is a typo. Please correct.

L90-91 – Would the authors provide a sentence to explain why POCs that appear to develop in response to gravity waves are excluded?

L117- 258 POCs seems to be a lot for a 3 month period, compared to 23 identified by Wood et al. (2008) over a two month period. Are these all separate POCs that form? How many of these merge with each other? Looking at Figure 9, I can potentially see how one can reach 258 POCs over a ~90 day period but cannot imagine a stratocumulus cloud deck with 20+ POCs embedded within it, especially since Figure 2 and 3 show just one POC. Would the authors show a snapshot with multiple POCs in one day so that we can see the shape and size of POCs that are identified on a frequent POC day, in addition to the classical cases shown in Figure 2 and 3?

L121 How sensitive are results if the 0.5degx0.5deg box is relaxed to something like 1x1 deg? Are all subsequent comparisons of variables (clouds, environment) averaged over the 0.5 deg x 0.5 deg box following the wind trajectory starting from initial formation? Or are they averaged over the whole POC?

L126-127 – how many POCs intersect a new POC? I assume this means that a POC merges with another POC? How often do these occur?

L155-156 - Are LWP and N derived during the nighttime? I haven't heard of such work before. Are there previous studies that have evaluated the validity of using nighttime optical depth and re retrievals to calculate LWP and N?

L159 – degree sign

Figure 4 – Why is the color scale cutoff at 10-2 mm/d-1 when the AMSR rain probability plot shows a 0.1 mm/d precipitation threshold?
And what does the right figure show? At 0.1 mm/d, we get a probability of ~2% and at 10 mm/d a probability of ~80%. Is it the probability of the rain rate being greater than the value along the x-axis? Are the rain rates area-averaged? Please provide some more explanation.

L190 – For calculating LTS, the surface AIR temperature should be used rather than the sea-surface temperature, since it's a measure of the atmospheric column instability.

L191 – Similar to the comment for L190, is the surface air temperature used or the sea-surface temperature?

L221 – What local times have previous studies found POCs to form at?

L233 – Figure 9, rather than Figure 11.

Figure 9 caption – median counts, rather than duration.

Fig 12 – there are large discontinuities along the day/night boundary in N, re, optical depth. How many of these are due to retrieval differences and how many from actual cloud changes?

L271 – I believe the authors mean reduced N, rather than elevated N.

L271-272 – How do the authors reconcile these observations with the rain rate plot, which seems to indicate similar rain rates? Can the authors comment further about this?

Figure 14 – The bins don't seem to overlap. Is there a reason for this? Would the authors provide more explanation for the choice to weight the frequency by the bin-mean rain rate? I can see that by weighting the distribution by rain rate, one can discern the difference in mean-precipitation rate between POC and nonPOC trajectories, but it makes it more difficult to ascertain the relative frequency of certain rain rates occurring in POC and nonPOC trajectories, especially before the formation of POCs (ie, asking whether exceeding a particular rain rate makes it X% more likely to form a POC).

L304-307- How good is MERRA2 reanalysis in capturing and characterizing boundary layer characteristics in POCs? I suspect that satellite retrievals of boundary layer characteristics are not well captured due to the large gradients. For example, does MERRA2 capture the boundary layer decoupling that is frequently observed in POCs? If not, I would be suspicious of the boundary layer output being compared.

L315 – "…have a maximum area larger than 104 sq km… " – The value here is likely wrong, given that the minimum size is 0.5degx0.5deg. However, I am curious how sensitive is this size to the 0.5x0.5deg minimum threshold used to identify POCs. I suspect not so much, but I am curious, given that the results here differ from Watson-Parris et al. (2021).

L317-318 – Precipitation is discussed as if it's solely driven by boundary layer dynamics and microphysics, but certain environmental conditions do make precipitation formation more conducive, as was discussed by Wood et al. (2008). LWP, re, and N appear to be different before POCs form (L271-272). Are there POC and nonPOC differences in Fig 10 or 15 that would be conducive to increasing LWP or decreasing N?

---

## Community Comment (CC2)

[Figure]

*Figure 1: Flowchart of how POCs are identified.*

[revised manuscript text omitted]

---

## Author Comment (AC1)

**Reply to Reviews**

**Reviewer comments are bolded,** *our answers are italicized*, any text in the manuscript that did not change is red, and modified text is blue

**General remarks to both reviewers:**

We thank the reviewers for their thorough examination of this manuscript, and we believe that we have now addressed your concerns. The primary major concerns of both reviews seemed to focus on: 1) better exploiting the high temporal resolution of the data 2) quantifying the growth rate of POCs, 3) inaccurate use of the term CONTROL, 4) insufficient description of uncertainty of cloud retrievals, and 5) more analysis of POC termination. Below we have extensive responses to all of these concerns. In short we have now 1) resolved cloud properties along the trajectories using the 10 minute data, 2) quantified growth rates of POCs, 3) eliminated use of the term CONTROL, 4) expanded the discussion around retrieval uncertainties, and 5) presented an analysis of the statistics of POCs of various termination fates (re-close vs. exit the StCu). Our detailed point by point responses to each reviewer comment follows.

**Reviewer 1 Comments**

**The paper demonstrates the value of high spatio-temporal resolution (10 min, 0.5 to 2 km) geostationary satellite data from GOES-16 for examining the lifecycle of Pockets of Open Cells (POCs) in the southeast Pacific. Most of the findings confirm previous work. Three areas need to be addressed: A) The new findings related to re-closing need more explanation. B) Retrievals of cloud properties from areas of broken cloud have high uncertainties, more relevant would be to emphasize comparisons of cloud properties in overcast conditions an hour or more *before* POCs form. C) Issue with "corresponding" POC and CONTROL tracks**

**A) New findings related to re-closing need more explanation**

> **The implication in the paper that marine stratocumulus boundary layer is well mixed all the time is not correct. There is a diurnal cycle in the degree of coupling between the ocean surface and cloud base with more coupled (well mixed) conditions overnight when negative buoyancy of parcels is strong compared to more decoupled conditions during the day (see Burleyson et al. 2013, esp. Fig 3 and 4) including situations during the day where there is a stable layer within the subcloud layer (Wilbanks et al. 2015). The diurnal cycle in the degree of coupling of the boundary layer means that there is a much lower probability that a POC that is present during the day will close compared to a POC at night.**

> **Based on Figure 7, most POC tracks end during the day.**

> *It is correct that the diurnal cycle in coupling of the ocean surface and cloud base will influence StCu cloud fractions which is possibly why most POC tracks end during the day. We have modified lines 236-242 to clarify this:* "*Why might POCs preferentially end (exit the StCu) during the day? We find that StCu area reaches a minimum around 12 local time (Figure s-1). As a result, we hypothesize that the tendency for POCs to exit the StCu during the day may simply be the result of a general reduction in StCu extent during the daylit hours (Burleyson and Yuter, 2015), so that the StCu edge effectively moves towards the POC during sunlight. This hypothesized mechanism is related to the tendency for the marine boundary layer to decouple from the surface during the daytime (e.g. Burleyson et al. 2013; Wilbanks et al. 2015) which also reduces the likelihood of POCs re-closing during the day.*"

**Further detail is needed regarding the statistics of POCs that do not reclose in order to distinguish among several scenarios. In particular, the size of the subset of POCs that essentially run off the cloud deck needs to be quantified Suggest add a table that addresses the number of tracked POCs and their median lifetimes, and median areas for the following categories:**

**POCS that form during the night and reclose POCS that form during the day and reclose**

**The category of "not reclose" includes**

**POCs that form at night and reach edge of cloud deck during the following day**

**POCS that form at night and do not reclose the following night**

**POCS that form during the day and reach edge of cloud deck before the next evening**

**POCS that form during the day and do not reclose the following night**

*We created Table R-1 to address this. For the never re-close cases, it shows that most POCs form at night and end during the day. The median duration ranges between 8 and 40 hours, however groups with the largest median duration typically contain the fewest number of cases. POC area ranges between 15000 and 56000 km², with the smallest POCs typically ending during the day, There may be a couple of reasons for the differences in POC area for POCs ending during the day compared to the night: 1) values may be skewed due to the changes in GOES-16 channels used to identify POCs, and 2) the small number of POCs that end at night (more likely reason). Most POCs that do re-close form during the day, have similar durations, and smaller areas to the POCs that never re-close. Visual evidence suggests that POCs that reclose may indeed be smaller, but it is difficult to make that conclusion based on between 1 to 4 cases in any given group. Considering only 12 cases re-close, we don't think analyzing both groups separately is best, because we don't necessarily think the statistics for the POCs that do re-close are reliable. A more thorough comparison of POCs that do and don't re-close is definitely interesting for the future, but many more months of data are likely needed which we don't have the time to process. Therefore, we added a version of Table R-1 (Table 1) to the manuscript which shows the different groups (e.g. forms at night ends during day, forms at night ends at night, etc.) for all cases that do and don't re-close. This table is now*

*referenced on lines 232-233 Referencing the table, we added lines 358-359:*
*"Table 1 shows that most POCs form at night and end during the day, and have a median duration typically between 10 and 30 hours." and lines 245-247: "POC area does not depend on when they start or end (Table 1) but tend to grow larger the longer they last, with POCs lasting > 30 hours having an average area approximately 3.6 times larger than POCs lasting < 10 hours."*

| Never Re-Close | | | |
|---|---|---|---|
| | | | |
| | # of cases | Median lifetime [hr] | Median Area [km$^2$] |
| Form at night and end during day | 76 | 12.6 | 15844 |
| Form at night and end at night | 9 | 30.7 | 18460 |
| Form at night and end at twilight | 6 | 8.9 | 24484 |
| Form at day and end during night | 4 | 18.3 | 28304 |
| Form at day and end during day | 23 | 22.2 | 16240 |
| Form at day and end during twilight | 3 | 39.3 | 17908 |
| Form at twilight and end during day | 5 | 26.2 | 18244 |
| Form at twilight and end during night | 2 | 11.0 | 56100 |
| Form at twilight and end during twilight | 1 | 36.2 | 17332 |
| | | | |
| Does Re-Close | | | |
| | # of cases | Median Lifetime [hr] | Median Area [km$^2$] |
| Form at night and end during day | 1 | 8.7 | 1992 |

| | | | |
|---|---|---|---|
| Form at night and end at night | 0 | n/a | n/a |
| Form at night and end at twilight | 2 | 8.8 | 788 |
| Form at day and end during night | 2 | 31.0 | 71464 |
| Form at day and end during day | 4 | 22.3 | 5406 |
| Form at day and end during twilight | 0 | n/a | n/a |
| Form at twilight and end during day | 3 | 38.9 | 7136 |
| Form at twilight and end during night | 0 | n/a | n/a |
| Form at twilight and end during twilight | 0 | n/a | n/a |
| | | | |
| **Overall Statistics** | | | |
| | # of cases | Median Lifetime [hr] | Median Area [km$^2$] |
| All Never Re-Close Cases | 129 | 16.3 | 17474 |
| All Does Re-Close Cases | 12 | 14.6 | 53692 |
| All Valid Cases | 141 | 16.33 | |

*Table R-1: Shows the total number of cases, median lifetime, and Median area for POCs that start/end during the day, night or twilight (defined as solar-zenith angles between 65o and 90o) for valid POC cases (those that follow a POC throughout its entire lifetime) for the cases the never and do re-close.*

**Additionally consider modifying Figure 5 to also indicate whether the end of POC track coincides with edge of the cloud deck.**

*A version of Figure 5 (Figure R-1 below) from the manuscript is shown below with the start and end times color coded by if a POC does or never re-close. It shows that there may be a slight westward shift in re-closing POC starting locations, but not much difference in the end locations. However, because of the small number of cases*

*following POCs that do re-close, it is difficult to make definitive conclusions about this. As a result, we don't include Figure R-1 in the paper.*

[Figure]

*Figure R-1: The start locations for the POCs that do re-close (red) and never re-close (blue) are shown in panel A), while the end locations for the POCs that do re-close (red) and never re-close (blue) are shown in panel B).*

**B) Retrievals of cloud properties from areas of broken cloud have high uncertainties**

**Coakley et al. 2005 showed that retrievals of cloud properties from areas of broken cloud have high uncertainties. Hence, comparisons between POC (which is by definition broken cloud) and CONTROL for cloud optical depth, cloud top effective radius, liquid water path, and cloud droplet number concentrations are problematic. Suggest refocus presentation of results to emphasize comparisons of the distributions of cloud properties of overcast clouds an hour or more before POC formation with overcast conditions that do not form POCs. This information may already be in the paper but needs to be clarified and emphasized.**

*We agree that the uncertainties of cloud property retrievals inside POCs may be problematic, however the primary reason we include the nighttime results - i.e. we expect that if we see similar patterns in both the daytime and nighttime results, we suspect that the changes found in the different cloud properties are real and not due to uncertainty resulting from broken cloud. To clarify this, we added lines 161-164:* *"Overall, uncertainties in microphysical retrievals inside POCs (Coakley et al. 2005) are unavoidable. However, the nighttime retrievals provide an independent comparison to the daytime retrievals that can lend confidence to the patterns, not magnitudes, in the daytime microphysical patterns and alleviate concerns about daytime retrievals due to broken cloud (Coakley et al. 2005)."*

**Since you have a Lagrangian track and run it up to 6 hours before POC development (Line 125), you can estimate the position of the air mass that forms the POC an hour or more prior to formation. Figure 13 shows "BEFORE" conditions but it is not clear in the paper what the increment of time before means other than to note it is before visible cloud transition. Is this 10 min before, 1 hour before, 2 hours before? Is it the median of the conditions 6 hours before? One might expect a larger LWP in overcast clouds that later form POCs since those clouds likely precipitate more but it is not obvious if other cloud characteristics would appreciably differ an hour or more before POC formation.**

*Regarding the "before" period - it ends 10 minutes before the cloud transition , and the "after" period represents 10 minutes after POC end. Lines 129-130 have been modified to clarify this: "The trajectories are run forward and backward from 10-minutes before POC development up to 6 hours before POC development and 6 hours after POC dissipation."*

*Regarding the 6-hours before POC formation, the dots shown in Figure 13 do represent the median conditions prior to POC development. Lines 314-316 have been modified to clarify this: "Further breaking this down into the overall statistics for the before, during, and after period, Figure 14 shows the distributions of the cloud field and precipitation characteristics of all POCs and their associated CLOSED trajectories represented by the median and $10^{th}$-$90^{th}$ percentile spread (see Figure S2 for the full daytime distributions)."*

*Regarding LWP prior to POC formation, I agree it isn't obvious from Figure 13 that LWP is higher prior to POC formation, however we did a new analysis, described below that shows that LWP is highest 6 hours before POC formation and continuously decreases through 10-hours after POC formation.*

**Also consider examining if there are any trends in cloud characteristics in the overcast clouds over the 6 hours prior to the POC opening up.**

*To address this, Figure R-2 shows the change in the median cloud property variables (cloud-optical depth, liquid-water path, cloud-top effective radius, and cloud-droplet number concentration) and cloud fraction, with the spread in the interquartile range overlaid (before the dashed line) and the grey region representing the first 10 hours of POC evolution. For the first six hours prior to POC start, we see that cloud optical depth decreases, liquid-water path generally decreases, cloud-top effective radius increases, and cloud-droplet number concentration decreases during the day relative to the CONTROL trajectories. As a result, we modified lines 301-321 to:*

*"One case gives us a glimpse at how cloud field properties vary throughout a POC's lifetime, and how they compare to a CLOSED trajectory. However, more cases are needed to make more definitive conclusions. For a more in-depth analysis, Figure 13a-e shows the trends of the cloud-field properties for the 6 hours before POC formation until 10 hours after formation for all POCs and the equivalent time-snippet for the CLOSED trajectories. Note that this is only daytime data due to retrieval differences during day and night (Section 2.3). The results shown are similar to Figure 12, with CF decreasing (Figure 13a), COD decreasing (Figure 13b), LWP decreasing (Figure 13c), $r_e$ increasing (Figure 13d), and N (Figure 13e) decreasing along the POC trajectories relative to the CLOSED trajectories. Increasing $r_e$ and decreasing N are consistent with an increased likelihood of precipitation (Stevens et al. 2005; Wood et al. 2008; Wood et al. 2011; Sarkar et al. 2019; Eastman et al. 2021 ). The overall differences in COD and LWP are not that large between the POC and CLOSED trajectories, however the slight decrease in LWP may be an indication of a loss of LWP due to precipitation flux in POCs. Interestingly, there is a bump in COD and LWP around 5 hours after POC development for the CLOSED trajectories. We suspect that this feature may result from the changing sample number shown in Figure 13f where the lowest number of samples (possibly due to cirrus contamination or time of day) occur just prior to this.*

*Further breaking this down into the overall statistics for the before, during, and after period, Figure 14 shows the distributions of the cloud field and precipitation characteristics of all POCs and their associated CLOSED trajectories represented by the median and $10^{th}$-$90^{th}$ percentile spread (see Figure S2 for the full daytime distributions). It confirms the differences in CF, $r_e$, and N are largest between the POC and CLOSED trajectories, with the most overlap in COD and LWP. The overall differences between the POC and CLOSED trajectories do not change much in the transition from the during-to-after periods, likely because most POCs never re-close before exiting the StCu. Finally, Table 2 shows the number of samples in the before, during, and after periods are sufficient to make robust conclusions, even though our study is limited to 3 months and the before and after periods are limited to at most 6 hours."*

[Figure]

*Figure R-2: The median changes in daytime cloud fraction (panel A), cloud optical depth (panel B), liquid water path (panel C), effective radius (panel D), and cloud drop number concentration (panel E) for the six hours before until ten hours after POC development are shown along all POC (red) and CLOSED (blue) trajectories, where the red and blue fill represent changes in the interquartile range. The number of valid samples for both the GOES-16 microphysical properties (purple) and GOES-16 cloud fraction (green) are shown in panel F). Gray shading indicates the time after POC development.*

*At night, Figure R-3 shows similar results as in the day (Figure R-1) with one main difference. The magnitudes of COD and LWP are smaller. This is because of the limitations of the GOES-16 nighttime algorithms, but overall the similar patterns to the daytime results increase our confidence that this result is real. Lines 322-334 have been modified to address this:*

*"We now make a similar comparison of POC and CLOSED trajectories with nighttime data. We caution that the magnitude of the night-time microphysical retrievals is not reliable due to the limitations in dynamic range, however the patterns in the night-time data can help confirm the validity of those seen in the daytime data, which is itself subject to biases that are associated with changes in cloud fraction. Figures S3 and 14 (see Figure S4 for the full nighttime distributions) shows that the overall patterns in CF, COD, $r_e$, N, and how they compare to the CLOSED trajectories are similar to the daytime characteristics of all POCs, with one main difference. The COD and derived LWP are substantially smaller than in the daytime data, and the CLOSED and POC LWP are essentially indistinguishable at night. This is likely an artifact of the limited dynamic range of the GOES-16 nighttime algorithms, resulting in inaccurate magnitudes. Nevertheless, given the substantial algorithmic differences in retrieved COD and $r_e$ at night versus during the day, it is encouraging that the microphysical patterns (i.e. $r_e$ and N) are similar. In particular, one might be concerned that the daytime differences between the POC and CLOSED trajectories are merely an artifact of 3D radiative transfer artifacts differentially affecting the two regimes (Zhang et al., 2012; Liang et al. 2015). However, the fact that these signals are also observed in the emission-based nighttime data lends credence to their sign, while uncertainty in their magnitude remains."*

[Figure]

*Figure R-3. The median changes in nighttime cloud fraction (panel A), cloud optical depth (panel B), liquid water path (panel C), effective radius (panel D), and cloud drop number concentration (panel E) for the six hours before until ten hours after POC development are shown along all POC (red) and CONTROL (blue) trajectories, where the red and blue fill represent changes in the interquartile range. The number of valid samples for both the GOES-16 microphysical properties (purple) and GOES-16 cloud fraction (green) are shown in panel F). Gray shading indicates the time after POC development.*

**Related to Figure 13-16, please add information to indicate sample sizes for BEFORE, DURING and AFTER statistics. Since most POCS run off the cloud deck presumably "AFTER" is only for the subset of POCs that reclose? Are sample sizes for BEFORE and DURING the same?**

*We have added Table 2 (Table R-2 below) to the manuscript showing the daytime sample sizes for the different products during the day, Table S-1 (Table R-3 below) to the supplement showing the nighttime sample sizes, and Table S-2 (Table R-4) to the*

supplement showing the environmental sample sizes. As shown, sampling, especially of the microphysical products, is not an issue. However, as you mentioned, the before and after (includes both POCs that do and don't reclose) have the lowest number of samples and the number of samples in both groups are not the same. To address this, we added lines 484-487: *"Finally, Table 2 shows the number of samples in the before, during, and after periods are sufficient to make robust conclusions, even though our study is limited to 3 months and the before and after periods are limited to at most 6 hours."*, added lines 319-321: *"Some of this spread may result in a much more limited number of AMSR-2 observations compared to GOES-16 (Tables 2 and S-1), but our results showing more intense precipitation for the POC compared to the CLOSED trajectories is consistent with expectations."*, and modified lines 343-345 *"Although the number of MERRA-2 samples relative to the other datasets are low (Table S-2), these results are consistent with the expectation that few differences exist between large-scale forcing where POCs form and those where they do not."*

| POC | | | |
|---|---|---|---|
| | Before | During | After |
| Cloud Fraction | 1451 | 5976 | 3434 |
| Microphysics | 239079 | 901451 | 461053 |
| Rain Rates | 4551 | 11810 | 8270 |
| CONTROL | | | |
| Cloud Fraction | 1321 | 5507 | 3125 |
| Microphysics | 273661 | 977695 | 335909 |
| Rain Rates | 3271 | 8378 | 6698 |
| | | | |

*Table R-2 : The number of Daytime samples for GOES-16 cloud Fraction, GOES-16 Microphysics (cloud-optical depth, liquid-water path, cloud-top effective radius, and cloud-drop number concentration), and AMSR-2 rain rates is shown.*

| POC | | | |
|---|---|---|---|
| | Before | During | After |
| Cloud Fraction | 2678 | 8524 | 857 |
| Microphysics | 426382 | 1208660 | 107205 |
| Rain Rates | 7211 | 33107 | 722 |
| CONROL | | | |
| Cloud Fraction | 2220 | 7494 | 711 |
| Microphysics | 356752 | 1437552 | 350815 |
| Rain Rates: | 11263 | 49663 | 1340 |

*Table R-3: The number of Nighttime samples for GOES-16 cloud Fraction, GOES-16 Microphysics (cloud-optical depth, liquid-water path, cloud-top effective radius, and cloud-drop number concentration), and AMSR-2 rain rates is shown.*

| Environmental Statistics | | |
|---|---|---|
| Before | During | After |
| 278 | 1161 | 263 |

*Table R-4 The number of MERRA-2 environmental samples for all variables and each time period is shown.*

**C) Issue with "corresponding" POC and CONTROL tracks.**

Calling the CONTROL track that is +/- 24 hours from the POC track but in the same start location the "corresponding CONTROL track" is a big stretch since the stratocumulus cloud deck itself is not steady state over 24 hour periods.

I do not find the line of reasoning Line 136 "to ensure that both the POC and CONTROL trajectories travel through similar meteorological regimes at

**the same time of day" very compelling given typically obvious changes in a southeast Pacific cloud deck and high pressure patterns in 24 hours. Suggest either justify this further or remove all materials related to "corresponding" tracks.**

**The CONTROL tracks are still relevant as a data set of conditions in overcast clouds. My objection is to the idea that a specific track within overcast cloud 24 hours before or after a POC corresponds directly to that specific POC (i.e. other than the fact that the POC occured all other key environmental conditions are equal).**

*We can't remove the comparison to the control trajectories, because that is the crux of our analysis. Specifically, we do not necessarily care if the state of the StCu along a POC trajectory is exactly equivalent to the state of the StCu 24 hours before or after a POC. Rather, we are only concerned with comparing a patch of stratocumulus that develops into a POC and one that does not. However we do agree that, just because the meteorology that both the POC and Control trajectories travel through may be similar, does not mean that the variables that control low-cloud fraction are exactly the same along both trajectories. To clarify this, we replaced CONTROL with CLOSED (approximate control) throughout the manuscript, and modified lines 139-143 to reflect this:* "To compare closed-cell StCu that develop into POCs and closed-cell StCu that do not, we run additional trajectories initialized at the same starting location of each POC trajectory at ±24-hour intervals until we find a comparison (CLOSED) trajectory that does not intersect any POCs to ensure both trajectories travel through similar meteorological regimes at the same time of day. Note, that this is only an approximate control because it is impossible to obtain a true control in which all variables that set CF are exactly equivalent."

Specific Comments:

**1. Line 90-91, mentions that POCS formed in response to gravity waves are excluded. How is this done?**

*This is done visually - i.e. I visually identify gravity waves, and if any POCs develop in response, I don't include them. To clarify this, Lines 90-93 have been modified to:* "For consistency, we classify any region of clearing completely enclosed within the StCu deck as a POC with the following conditions: 1) regions of clearing at the StCu edge that become completely enclosed within the StCu deck are not classified as POCs, and 2) any potential POCs that we visually identify to develop in response to gravity waves (Allen et al. 2013) are not included."

**2. Section 2.4, Please clarify that AMSR-2 data used for the precipitation estimation is available only at most twice a day (swath width means revisit time is every 2 days) and thus provides the equivalent of a few minutes information along the median 10 hour POC duration.**
**[https://www.ospo.noaa.gov/Products/atmosphere/gpds/about_amsr2.html]**

*We added lines 190-192 to address this: "Note, because AMSR-2 is polar orbiting and only available at most twice a day, these co-locations between AMSR-2 and GOES-16 only provide limited snapshots of precipitation along each trajectory, typically at times near the equator crossing time of the A-Train satellite constellation, at 0130 and 1330 local time."*

**3. Section 3.2, Please provide information that quantifies POC occurrence normalized by the area of the overall cloud deck in the study region. To what degree are fewer POCs occurring because there is less cloud deck area? This is an important constraint on understanding environmental controls on POC formation. Based on Figure 10, the area with cloud fraction > 80% is substantially larger for days with >= 7 POCs compared to those with 0 POCS. Related to this, it also would be helpful to add the time series of daily Sc cloud area to Figure 9.**

*To address both of these issues, I have modified Figure 11 (Figure R-4 below) to show the 10-minute time series in both stratocumulus area (red) and number of cases (blue). It shows that whan stratocumulus are large, there are typically larger numbers of POCs (e.g. November 1 - November 15). We added line 423 to address this: "Figure 11a shows that StCu area is typically larger when there are more POCs."*

[Figure]

*Figure R-4: Changes in stratocumulus (StCu determined using 0.5-2-km level 1.5 observations) and number of POCs every 10 minutes are showin in Panel A), and the distributions of [MOU1] area for days with no POCs (blue) and days with > 7 POCs (brown) are shown in Panel B).*

**4. Figure 11 caption, please clarify what is the data set used to compute Sc cloud area. Is this the 0.5 - 2 km pixels from the GOES-16 sensor.**

Yes, this is discussed in the POC Identification section of the manuscript, but I can add this clarification to the Figure 11 caption to: *"Changes in stratocumulus (StCu determined using 0.5-2-km level 1.5 observations) and number of POCs every 10 minutes are shown in Panel A), and the distributions of area for days with no POCs (blue) and days with > 7 POCs (brown) are shown in Panel B)."*

**5. Figure 12 caption discussed purple and green dots but the actual figure shows blue and red dots.**

*The Figure 12 caption has been changed to: "Changes in cloud fraction (panel A), cloud optical depth (panel B), liquid water path (panel C), effective radius (panel D), and cloud drop number concentration (panel E) are shown along the trajectory for a sample POC case that occurred between 09/03/2019 and 09/07/2019. The red dots correspond to*

the sample POC trajectory, while the blue dots correspond to the corresponding CONTROL trajectory. Gray shading indicates night time periods."

**Reviewer 2 Comments:**

**Review of Smalley et al "A Lagrangian Analysis of Pockets of Open Cells over the Southeast Pacific"**

**The manuscript provides an observational analysis of pockets of open cells (POCs) over the southeast Pacific during a three-month period using retrievals from the geostationary satellite GOES16 and other satellite/reanalysis products. Once the POCs are identified, the authors use MERRA2 reanalysis winds to track how their size and cloud and environmental properties change during the POC evolution, from before its formation to after its demise. The authors also run trajectories that start 24 hours before or after initial POC formation in cloudy overcast scenes to compare how the POC trajectories differ from a "control" trajectory case where no POCs form.**

**Although there have been a couple previously published, satellite-based analysis of POCs (Wood et al. 2008; Watson-Parris et al., 2021), the novel aspect of this study is the use of Lagrangian trajectories to examine the temporal aspect of clouds, particularly how they form and how they dissipate. Information of their temporal duration and area are new and interesting. However, I found that much of the manuscript focuses on the results that have been confirmed by past observations (more rain rate, larger cloud drops, lower aerosol or cloud droplet number concentration in POCs) rather than highlighting the new findings and its implications. I believe that the authors can make significant contributions to answering what conditions favor POC formation and what sets the size of POCs, which other observational studies have not been able to discuss at length. In the current manuscript, the discussion of these results seems minimal. In addition, there are a number of errors and issues associated with the methods and points of clarification that I would like to see the authors address.**

**Given that some of the methodological fixes potentially impact results and interpretations, I recommend major revisions before the manuscript is published.**

**Major comments**

**Lack of emphasis and discussion on the temporal evolution of POCs**

> **The authors point out the novelty of their analysis is in capturing the temporal evolution of POC formation, maintenance, and dissipation. And based on the title, I expected to see more emphasis and discussion on such themes. However, most of the results of the paper seemed to focus on confirming past findings about the POC vs non-POC cloud and**

**environmental conditions that have already been reported by previous studies, which did not use a Lagrangian perspective. That POCs have less clouds, larger cloud droplets, less aerosol/cloud drops, more intense rain rates, and that they have similar large-scale meteorological contexts as non-POC regions have been reported previously by examining POC conditions and the regions surrounding POCs, without looking at a Lagrangian perspective.**

Lagrangian Look at how Cloud Properties:

*To address this, Figure R-2 shows the change in the median cloud property variables (cloud-optical depth, liquid-water path, cloud-top effective radius, and cloud-droplet number concentration) and cloud fraction, with the spread in the interquartile range overlaid (before the dashed line) and the grey region representing the first 10 hours of POC evolution. For the first six hours prior to POC start, we see that cloud optical depth decreases, liquid-water path generally decreases, cloud-top effective radius increases, and cloud-droplet number concentration decreases during the day relative to the CONTROL trajectories. As a result, we modified lines 301-321 to:*

*"One case gives us a glimpse at how cloud field properties vary throughout a POC's lifetime, and how they compare to a CLOSED trajectory. However, more cases are needed to make more definitive conclusions.* *For a more in-depth analysis, Figure 13a-e shows the trends of the cloud-field properties for the 6 hours before POC formation until 10 hours after formation for all POCs and the equivalent time-snippet for the CLOSED trajectories. Note that this is only daytime data due to retrieval differences during day and night (Section 2.3). The results shown are similar to Figure 12, with CF decreasing (Figure 13a), COD decreasing (Figure 13b), LWP decreasing (Figure 13c), $r_e$ increasing (Figure 13d), and N (Figure 13e) decreasing along the POC trajectories relative to the CLOSED trajectories. Increasing $r_e$ and decreasing N are consistent with an increased likelihood of precipitation (Stevens et al. 2005; Wood et al. 2008; Wood et al. 2011; Eastman et al. 2019; Sarkar et al. 2019). The overall differences in COD and LWP are not that large between the POC and CLOSED trajectories, however the slight decrease in LWP may be an indication of a loss of LWP due to precipitation flux in POCs. Interestingly, there is a bump in COD and LWP around 5 hours after POC development for the CLOSED trajectories. We suspect that this feature may result from the changing sample number shown in Figure 13f where the lowest number of samples (possibly due to cirrus contamination or time of day) occur just prior to this.*
*Further breaking this down into the overall statistics for the before, during, and after period, Figure 14 shows the distributions of the cloud field and precipitation characteristics of all POCs and their associated* *CLOSED trajectories represented by* *the median and $10^{th}$-$90^{th}$ percentile spread (see Figure S2 for the full daytime distributions). It confirms the differences in CF, $r_e$, and N are largest between the POC*

*and CLOSED trajectories, with the most overlap in COD and LWP. The overall differences between the POC and CLOSED trajectories do not change much in the transition from the during to after periods, likely because most POCs never re-close before exiting the StCu. Finally, Table 2 shows the number of samples in the before, during, and after periods are sufficient to make robust conclusions, even though our study is limited to 3 months and the before and after periods are limited to at most 6 hours."*

*At night, Figure R-3 shows similar results as in the day (Figure R-1) with one main difference. The magnitudes of COD and LWP are smaller. This is because of the limitations of the GOES-16 nighttime algorithms, but overall the similar patterns to the daytime results increase our confidence that this result is real. Lines 322-334 have been modified to address this:*

*"We now make a similar comparison of POC and CLOSED trajectories with nighttime data. We caution that the magnitude of the night-time microphysical retrievals is not reliable due to the limitations in dynamic range, however the patterns in the night-time data can help confirm the validity of those seen in the daytime data, which is itself subject to biases that are associated with changes in cloud fraction. Figures S3 and 14 (see Figure S4 for the full nighttime distributions) shows that the overall patterns in CF, COD, $r_e$, N, and how they compare to the CLOSED trajectories are similar to the daytime characteristics of all POCs, with one main difference. The COD and derived LWP are substantially smaller than in the daytime data, and the CLOSED and POC LWP are essentially indistinguishable at night. This is likely an artifact of the limited dynamic range of the GOES-16 nighttime algorithms, resulting in inaccurate magnitudes. Nevertheless, given the substantial algorithmic differences in retrieved COD and $r_e$ at night versus during the day, it is encouraging that the microphysical patterns (i.e. $r_e$ and N) are similar. In particular, one might be concerned that the daytime differences between the POC and CLOSED trajectories are merely an artifact of 3D radiative transfer artifacts differentially affecting the two regimes (Zhang et al., 2012; Liang et al. 2015). However, the fact that these signals are also observed in the emission-based nighttime data lends credence to their sign, while uncertainty in their magnitude remains."*

**POC Growth:**

- *See Major Comment 4*

**I believe that the authors are however well positioned and should focus on answering some questions that other studies have only lightly touched on or have not addressed. First, they can better answer the origins of POCs and whether there are any environmental precursors that make it much more likely for POCs to form. For example, are there necessary conditions for POC formation?**

**The authors touch on this in Figure 13 and 14, and mention that there appear to be no environmental precursors to POC formation, but do not elaborate further about what it means with respect to what previous studies have found (e.g., Wood et al., 2008; Yamaguchi and Feingold, 2015) and whether the authors find evidence to support or go against what has been proposed in those studies.**

*This is the whole point of the paper that there are no environmental precursors to POC development, and that their development is entirely dictated by precipitation and corresponding changes in cloud microphysics. Figure R-5 further demonstrates this showing that there is very little difference between the different environmental variables along the POC and CONTROL trajectories. Most studies have made similar conclusions regarding the environment, with one exception. Studies have found that POC development may be influenced by off-shore flow by introducing aerosols. However, we found little difference in aerosols between the POC and CONTROL cases. Lines 367-376 were modified to clarify this:* *"Throughout a POC's lifecycle, similar environments are consistent with observational results showing POC occurrence does not depend on free-tropospheric moisture (Wood et al. 2008), and modeling (Berner et a. 2011) and observational (Sharon et al. 2006; Wood et al. 2011) results found little difference in moisture and potential temperature profiles between POCs and surrounding StCu. Prior studies have noted that off-shore flow from SEPAC may suppress open-cell development by introducing more aerosols into the environment (e.g. Wood et al. 2008; Abel et al. 2020), but our results indicate no aerosol difference between both sets of trajectories (Figure 16j-l) suggesting our trajectories originated over ocean. Although the number of MERRA-2 samples relative to the other datasets are low (Table S-2),* *these results are consistent with the expectation that few differences exist between large-scale forcing where POCs form and those where they do not. This implies that the process relevant to the formation and maintenance of POCs are small-scale processes within the PBL."*

[Figure]

*Figure R-5: The median changes in MERRA-2 boundary-layer height (panel A), boundary-layer water-vapor mixing ratio (panel B), lifted-condensation-level height (panel C), aerosol-optical depth (panel D), estimated-inversion strength (panel E), 700-mb water-vapor mixing ratio (panel F), and 700-mb omega (panel G) for the six hours before until ten hours after POC development are shown along all POC (red) and CLOSED (blue) trajectories, where the red and blue fill represent changes in the interquartile range.*

**The authors are also well positioned to answer the downsides of relying on polar orbiting satellites to identify and characterize POCs. Do we miss anything by just observing POCs twice a day, except for the lack of temporally tracking their evolution?**

*The primary limitations of using polar-orbiting satellites is not being able to capture the diurnal characteristics of POCs and the temporal variance in POC characteristics. Therefore, we added lines 378-380 to address this:* "This study develops a novel methodology to identify POCs and then uses a Lagrangian analysis to track their evolution and how that equates to comparison trajectories of closed-cell StCu that never transition, and expands upon prior polar-orbiting-studies that were unable to analyze the temporal evolution of POC characteristics."

**The other set of questions that the authors are able to address relates to the growth of POCs. For example, what sets the growth of POC size? And for larger POCs, do they grow rapidly or steadily over time? Are POC sizes just determined by how long it takes for the POCs from formation to reach the edge of the stratocumulus deck? When POCs grow, do they have a preferred direction of growth relative to the 925hPa trajectory?**

*Going into more depth regarding POC growth is a suggestion that we should have done in the initial draft. We find that, although most POCs are large (Figure R-6a and appear to grow larger the longer they last (Figure R-6b), POC mergers have a large impact on this. Therefore to realistically represent growth rate, we went through each POC trajectory and removed times when a merger occurs, or twilight causes error in the POC identification. After doing this, we found that most POCs have a positive growth rate (Figure R-6c), and generally have a steady growth rate no matter how long lived they are. For more detail, we modified lines 244-259 and modified Figure 8 in the manuscript:*

"Figure 8a shows the relative frequency distribution of median POC area, with values typically reaching $4.6 \times 10^3$-$4.7 \times 10^4$ $km^2$. Figure 8b shows changes in POC median area as a function of POC duration. POC area does not depend on when they start or end (Table 1) but tend to grow larger the longer they last, with POCs lasting > 30 hours having an average area approximately 3.6 times larger than POCs lasting < 10 hours. We could conclude that longer-lived POCs have more time to grow larger. However, 76% of distinct POCs that form end up merging at least once, which also contributes to increase POC area over time. Figure 8c shows the relative frequency of average POC growth rate after visually removing mergers, which cause artificial spikes in the growth rate. The median POC growth rate is 133 $km^2$ $Hour^{-1}$ with interquartile range between 42 and 422 $km^2$ $Hour^{-1}$. Figure 8d shows that the shortest-lived POCs occasionally grow faster than the longest-lived POCs, however the medians are not notably different. Furthermore, the results for smaller POCs may be biased due to limited sampling after removing mergers, problems during twilight, and cirrus contamination. We conclude that POCs tend to grow at a fairly consistent rate throughout their lifetime with significant departures from those rates associated with POC mergers."

[Figure]

*Figure R-6: The Histogram of POC median area is shown in panel A), the distribution of median POC area as a function of POC duration is shown as box plots in panel B), where the solid line represents the median duration, the Histogram of POC median growth rate is shown in panel C), and the distribution of median POC growth rate as a function of POC duration is shown as box plots in panel B). Note that in the boxplots, the shaded boxes represent the interquartile range, the solid lines inside the shaded regions represent the median, and the whiskers represent data between the 10th and 90th percentiles.*

*We also investigated if there is a preferred direction of POC growth relative to the 925-hPa wind vector. To do this, we sliced each POC along the wind vector (as a proxy for poc length) and perpendicular to the wind vector (a proxy for POC width) and used the average POC length over POC width as a function of time to test if there is a*

*preferred direction of growth. Figure R-7 shows the average change in both the along- and across-wind-vector lengths for all POCs. The average along-wind vector length is typically larger than the across-wind vector length at any given time. However, the 2σ spread in both variables at any given time is large and overlaps. Therefore, it suggests that there is not a preferred direction of growth relative to the wind vector, because both lengths increase at similar rates, and we do not include this result in the new manuscript as a result.*

[Figure]

*Figure R-7: The average along-wind-vector (red) and across-wind-vector (blue) lengths for all POCs as a function of time are shown. The 2σ spread is shown as the shaded regions for the along-wind-vector (light red), and across-wind-vector (light blue) distributions at any given time.*

**LTS & LCL calculations**

**For the LCL and LTS calculations, the surface air temperature is typically used, rather than the surface temperature (Romps, 2017; Klein and Hartmann, 1993). For the LTS, it seems that the sea surface temperature is used and potentially also for the LCL. I suggest the authors redo the analysis for these. Based on previous studies that have shown surface air temperatures tend to be colder in POCs due to precipitation evaporation, I would expect a lower LCL and stronger LTS over POCs vs non-POCs.**

*This was just a typo in the manuscript, because MERRA-2 2-meter temperature was used in the calculation of LTS. Lines 322 and 324 have been modified to address this as "2-m temperature".*

**Furthermore, my impression from the reading the manuscript is that the boundary layer qv and LCL are frequently discussed as environmental factors (alongside large-scale subsidence or sea level pressure differences), when internal cloud and boundary layer processes (precipitation and boundary layer decoupling) can have significant impact on these variables. Please provide further explanation for why the authors consider them as environmental factors mainly determined by the large-scale meteorology, rather than internal boundary layer processes.**

*I agree with the reviewer that the boundary-layer characteristics (e.g. boundary-layer qv and LCL height) depend on changes in cloud-scale processes. However, We doubt that these interactions are well represented in MERRA-2 because it struggles to constrain these interactions. Therefore, lines 361-366 have been added to address this:* *"Results from Sharon et al. (2006) are similar to ours showing that PBL depth and winds are similar in both open- and closed-cell environments, however they found the open-cell PBL is moister. This is likely because changes in cloud microphysics and PBL turbulence interact with the environment changing the moisture structure in the cloud layer, and the aircraft measurements Sharon et al. (2006) used likely captured these differences. The MERRA-2 PBL characteristics we show are likely more related to the large-scale environment, because small-scale processes are not well constrained in MERRA-2 (e.g. Witte et al. 2021)."*

Analyses in Section 3.2

**The synoptic comparison between no-POC and frequent (>7) POC days is interesting, but how robust are the results? In other words, how similar are the synoptics among frequent POC days vs no POC days compared to the difference between frequent and no POC days? There is no discussion about how large the POC vs non-POC differences are compared to the variance within each category.**

**It was also not clear to me what specific question the authors want to answer by examining the synoptic and environmental conditions. My impression is that the purpose of this section is to ask whether there are certain environmental conditions that favor POCs vs just common stratocumulus clouds. The statement at the end of the section (L250-254), which points out that many of the synoptic differences are consistent with conditions that favor more stratocumulus seems to muddy whether those synoptic differences we observe in the difference plots are indeed conducive for POCs specifically. Can the authors perform more in depth analysis with the available data (or by including other months) to solely**

**compare non-POC days when there are large StCu cloud decks to see whether there are certain conditions that actually favor POC formation?**

*For this study, adding more cases would be a significant time sink, and therefore this is an area for future research. However, we did analyze how the difference in means between the days with many POCs and those with no POCs compares to the overall variance of both groups by dividing the mean difference by the average standard deviation between both groups. If this ratio is large, this could mean two things: 1) the means are similar or 2) the standard deviations are small. Figure R-8 shows that this ratio is largest where the mean differences are largest (Figure 10 from the manuscript) for most variables (i.e. SLP, EIS, wind, PBL Height and moisture, and LCL height. For the manuscript, we calculated an average ratio for each characteristic weighted by cloud fraction. The average ratio was largest for the PBL characteristics, suggesting that differences between these distributions on days with no and many POCs are robust. However, acknowledge that there is no clear signal regarding which conditions favor the development of a larger number of POCs due to the limited sample size other than those influencing StCu amount. Lines 280-290 have been added To caveat this:*

*"It is reasonable to be concerned about the robustness of these results, because of the limited MERRA-2 sample size during days with no and many POCs. To quantify the physical significance of the differences in mean environmental properties, we calculated a signal-to-noise ratio as $\frac{\left(\mu_{many}-\mu_{none}\right)}{\frac{1}{2}(\sigma_{many}+\sigma_{none})}$, where $\mu_{many}$ and $\sigma_{many}$ represent the mean and standard deviation of each distribution on days with > 7 POCs and $\mu_{none}$ and $\sigma_{none}$ represent the same statistics on days with no POCs. We then calculate a regional mean of this quantity, weighted by average low-CF, for each environmental characteristic. This value ranges from 0.14-0.42 and is largest for the characteristics with the largest mean differences between days with no and many POCs (PBL height, water vapor, and LCL height), suggesting that the mean differences in these characteristics are the most physically meaningful. However, we note that there is in general a great deal of overlap between the distributions of all of the environmental characteristics on the days with large numbers of POCs and the no-POC days. Overall, it is unclear what environmental conditions, outside of those favoring StCu development, favor POC development due to limited sample size of days with few and many POCs. This is an area for a future investigation, which will require substantial increases in sample size."*

[Figure]

*Figure R-8: The ratio between $(\mu_2 - \mu_2)\,(\sigma_2 + \sigma_1)^{-1}$ where $\mu_2$ and $\sigma_2$ represent the mean and standard deviation for sea-level pressure (Panel A), estimated-inversion strength (Panel B), 50-meter winds (Panel C), PBL Height (Panel D), PBL water vapor mixing ratio (Panel E), LCL Height (Panel F), 700-mb $\omega$ (Panel G), and aerosol-optical depth (Panel H) on days with many POCs and $\mu_1$ and $\sigma_1$ represent the same variables for days with no POCs.*

**Note also that some things like, lower AOD, lower LCL, and higher qv might be a symptom of, rather than a cause behind having many POCs during a day, because POCs are cleaner, have lower LCL, and moisture near surface values compared to non-POC boundary layers.**

*We agree that differences in AOD and PBL characteristics are likely influence by cleaner air inside POCs, and have lines 272-274 to clarify this:* *"Furthermore, the differences between AOD, LCL height, and $q_v$ may be a symptom of a higher number of POCs, because POCs are cleaner, have lower LCLs, and more moisture near the surface compared to non-POC boundary layers."*

Specific/minor questions

**L8 – Although it also shows up in the conclusions, I believe that 104km2 is a typo. Please correct.**

*$10^4$ $km^2$ is now correct in the abstract.*

**L90-91 – Would the authors provide a sentence to explain why POCs that appear to develop in response to gravity waves are excluded?**

*We added lines 160-162 to address this: "POC development due to gravity waves are excluded because they close very quickly and move in the direction of wave propagation instead of with the mean wind."*

**L117- 258 POCs seems to be a lot for a 3 month period, compared to 23 identified by Wood et al. (2008) over a two month period. Are these all separate POCs that form? How many of these merge with each other? Looking at Figure 9, I can potentially see how one can reach 258 POCs over a ~90 day period but cannot imagine a stratocumulus cloud deck with 20+ POCs embedded within it, especially since Figure 2 and 3 show just one POC. Would the authors show a snapshot with multiple POCs in one day so that we can see the shape and size of POCs that are identified on a frequent POC day, in addition to the classical cases shown in Figure 2 and 3?**

*These are all separate POCs that form, but ~70% of the POCs identified merge together during their lifecycles. Currently, if two distinct POCs (e.g. POC A and POC B), they are still classified as separate POCs after the merger instead of a POC C or continuing one POC while ending the other. We didn't think this is an issue because even if the POCs do merge the individual trajectories would be sampling different regions of the POCs. We added line 248 to clarify this: "However, 76% of distinct POCs that form end up merging at least once, which also contributes to increase POC area over time."*

*This is primarily why you see days with a "large" number of POCs (i.e. many small POCs will form then begin to merge prior to exiting the cloud deck). An example of this is shown in Figure R-9 below, which shows many distinct POCs clustered primarily in two different regions (yellow ovals).*

[Figure]

2019-11-05 09:00

Figure R-9: This shows a night where many POCs develop and start to merge.

**L121 How sensitive are results if the 0.5degx0.5deg box is relaxed to something like 1x1 deg? Are all subsequent comparisons of variables (clouds, environment) averaged over the 0.5 deg x 0.5 deg box following the wind trajectory starting from initial formation? Or are they averaged over the whole POC?**

*The environmental variables are not averaged at all along each trajectory. Instead each environmental property is determined by mapping the nearest MERRA-2 gridpoint to each trajectory point. To make this explicit, We modified lines 149-152 to:* "We classify the large-scale environment at each point along the POC and CLOSED trajectories using sea-level pressure (SLP), estimated-inversion strength (EIS), 700-mb water vapor mixing ratio ($q_v$), 700-mb omega ($\omega$), planetary-boundary layer (PBL) height, PBL mean $q_v$, lifted condensation level (LCL) height, aerosol-optical depth (AOD), 50-m winds, and 925-mb wind speed/direction, all of which are derived from the nearest MERRA-2 grid point."

*The cloud properties for all GOES-16 pixels within each 0.5x0.5 deg box surrounding each trajectory point are all used in the composite analyses, and there little change in the composite statistics after widening our window to 1°x1° (Figure R-10). We modified lines 149-152 to address this: To address this in the manuscript, we changed the following sentence* *"We compare the following cloud properties along the POC and* *CLOSED* *trajectories: CF, cloud optical depth (COD), cloud top effective radius ($r_e$), liquid water path (LWP), and cloud drop number concentration (N). COD, $r_e$, LWP, and N are composited from cloudy pixels within a 0.5° x 0.5° window surrounding each trajectory point because this window size is close to the same size as a MERRA-2 gridbox.* *We find our results are not sensitive to window size."*

[Figure]

*Figure R-10: Daytime statistics for cloud fraction (A, B, and C), cloud-optical depth (D, E, and f), liquid-water path (G, H, and I), cloud-top effective radius (J, K, and L), cloud-drop number concentration (M, N, and O), and AMSR-2 rain rate (P, Q, and R) are shown in the upper-white half of each panel, while the nighttime statistics are shown in the grey-lower half of each panel. Red values represent the POC trajectories, while the blue values represent the CLOSED trajectories. The colored dots represent the median of each distribution, and the horizontal lines represent the spread between the 10th and 90th percentiles.*

**L126-127 – how many POCs intersect a new POC?I assume this means that a POC merges with another POC? How often do these occur?**

*We found that only 25 POC trajectories intersect a different POC in the before and after periods (54 out of 768 total pre-POC hours and 40 out of 782 post-POC hours), therefore we do not believe this has a large impact on our results. We added lines 132-133 to clarify this:* *"We do not expect this to substantially impact our results, because it represents only 7% of all pre-POC and 5% of all post-POC hours."*

*Additionally, this does not mean that mergers are occuring, rather we eliminate times in the before and after periods where the trajectories intersect another POC so that our cloud property analysis is correct.*

**L155-156 - Are LWP and N derived during the nighttime? I haven't heard of such work before. Are there previous studies that have evaluated the validity of using nighttime optical depth and re retrievals to calculate LWP and N?**

*Yes, LWP and N are derived at night, and, because of the differences in day and night retrievals of COD and re, we find the magnitude of LWP and N are smaller at night than during the day. However, we have not found any studies that have evaluated the validity of deriving LWP and N from nighttime retrievals.*

**L159 – degree sign**

*This typo on line 125 has been corrected to* *"0.5° x 0.5°"*

**Why is the color scale cutoff at 10-2 mm/d-1 when the AMSR rain probability plot shows a 0.1 mm/d precipitation threshold?**

*The values shown in the spatial plot were in mm/hour not mm/day. This is now correct and is shown in an updated version of* *Figure 4* *and Figure R-11 below.*

[Figure]

*Figure R-11 : AMSR-2 rain rates matched to the POC shown in Figure 2 overlaid on GOES-16 0.64-µm reflectance are shown in panel A). AMSR-2 rain probability as a function of rain rate for all September-November 2019 data over the entire domain are shown in panel B), where the solid black line represents the median probability at a given rain rate, grey fill represents the 10th-90th percentile spread at a given rain rate, and the dashed red line represents the rain rate threshold of 0.1 mm day-1.*

**And what does the right figure show? At 0.1 mm/d, we get a probability of ~2% and at 10 mm/d a probability of ~80%. Is it the probability of the rain rate being greater than the value along the x-axis? Are the rain rates area-averaged? Please provide some more explanation.**

*Regarding the first question, the probability represents the likelihood of rain reaching the surface. Lines 186-188 have been modified to clarify this:* "*To correct this, we use a precipitation threshold of 0.1 mm day$^{-1}$, which corresponds to t*he likelyhood of rain reaching the surface *typically below 5% (Figure 4b) and is consistent with the minimum observable value of surface rain (*Comstock et al., 2004*;* Rapp et al. 2013*).*

*Regarding the second question, the rain rates used are area-averaged. Lines 195-198 have been modified to clarify this:* "*These statistics are used to derive* the likelihood of precipitation and area-averaged*-CloudSat-like precipitation across the microwave radiometer swath, which allows for significantly more potential overlap with GOES-16 identified POC cases than does CloudSat.*"

**L190 – For calculating LTS, the surface AIR temperature should be used rather than the sea- surface temperature, since it's a measure of the atmospheric column instability.**

*This has been modified on line 207 to* *"2-m temperature".*

**L191 – Similar to the comment for L190, is the surface air temperature used or the sea-surface temperature?**

*This has been modified on lines 206-207 to* *"2-m temperature".*

**L221 – What local times have previous studies found POCs to form at?**

*Wood et al. (2008) found that POCs typically form between 0 and 6 local time (peaking around 3 local), similar to our results. Therefore, lines 233-236 have been modified to:* *"Further demonstrating this, Figure 7 shows the* *diurnal cycle of the relative frequency of POC start and end times, with POCs typically forming at night* *between 0 - 6 local time and ending during the day* *between 9 and 13 local time. This is consistent with* *findings by Wood et al. (2008) that showed POC formation peaks around 3 local time* *when precipitation is most intense and StCu are thickest (Wood et al. 2008; Burleyson et al. 2013)."*

**L233 – Figure 9, rather than Figure 11.**

*This typo on line 257 has been changed to* *"Figure 9".*

**Figure 9 caption – median counts, rather than duration**.

*The figure 9 caption now reads as:* *"The number of POCs occurring on each day is shown. The boxplot represents the overall distribution of daily POC occurrence, where the solid line represents the* *median counts**, the shaded box represents the interquartile range, and the whiskers represent data between the 10th and 90th percentiles."*

**Fig 12 – there are large discontinuities along the day/night boundary in N, re, optical depth. How many of these are due to retrieval differences and how many from actual cloud changes?**

*This is entirely due to the limitations in the nighttime retrievals of optical depth and effective radius, as well as each time series still showing some problematic pixels at twilight. Lines 323-329 have been modified to address this:* *"This is likely an artifact of the limited dynamic range of the GOES-16 nighttime algorithms,* *resulting in inaccurate magnitudes."*

**L271 – I believe the authors mean reduced N, rather than elevated N.**

*This has been removed from the manuscript after re-working section 3.3*

**L271-272 – How do the authors reconcile these observations with the rain rate plot, which seems to indicate similar rain rates? Can the authors comment further about this?**

*Yes, the bulk of the rain in both the CONTROL and POC cases are light as stated on lines 515-517: "Precipitation rates are typically low with median values between 0.14 and 0.44 mm day$^{-1}$ and they remain relatively constant from before POC development until after POC dissipation."*

*However, there is a longer and heavier tail of more intense precipitation in the POC cases. We added lines 342-345 to clarify this: "Notably, there is a larger spread in rain rate. Some of this spread may result in a much more limited number of AMSR-2 observations compared to GOES-16 (Tables 2 and S-1), but our results showing more intense precipitation for the POC compared to the CLOSED trajectories is consistent with expectations."*

*This indicates that, even though most of the rain is light, heavier rain is more frequent in POCs than the closed cells they are compared to. This is precisely why we show Figure 14. We changed lines 346-349 to clarify this: "The precipitation is usually light for both the POC and CLOSED trajectories (Figure 14p-r), Figure 14 clearly highlights that there is substantially more precipitation accumulation for the POC then the CLOSED trajectories during the POC lifetime."*

**Figure 14 – The bins don't seem to overlap. Is there a reason for this? Would the authors provide more explanation for the choice to weight the frequency by the bin-mean rain rate? I can see that by weighting the distribution by rain rate, one can discern the difference in mean-precipitation rate between POC and nonPOC trajectories, but it makes it more difficult to ascertain the relative frequency of certain rain rates occurring in POC and nonPOC trajectories, especially before the formation of POCs (ie, asking whether exceeding a particular rain rate makes it X% more likely to form a POC).**

*The bins for the POC and CONTROL cases don't overlap because we used a different number of bins for both cases due to sampling issues, and we chose the number of bins to create relatively continuous distributions.*

*The reason that we weight each PDF by bin-mean rain rate is to emphasize the accumulated rain. By doing this, we clearly show that heavier rain is more frequent/likely*

*inside a POC compared to general closed cells. Lines 336-339 explains this:* "Figure 15 shows the distribution of rate-weighted rain rate. Specifically, each bin is multiplied by bin-center rain rate, which places a higher weight on bins with more intense precipitation such that the area under the histogram is equal to the accumulated rainfall.".

*The unweighted distributions are shown in the 3rd supplemental figure, and they show that most precipitation is light for both groups, but the tail of the POC distributions is heavier, which is more subtle than Figure 14, but tells a similar story.*

**L304-307- How good is MERRA2 reanalysis in capturing and characterizing boundary layer characteristics in POCs? I suspect that satellite retrievals of boundary layer characteristics are not well captured due to the large gradients. For example, does MERRA2 capture the boundary layer decoupling that is frequently observed in POCs? - If not, I would be suspicious of the boundary layer output being compared.**

*MERRA-2 likely struggles in capturing and characterizing boundary layer characteristics in POCs, because it struggles to constrain the small-scale processes due to precipitation and radiative interactions. Therefore, we added lines 364-366 to address this:* "The MERRA-2 PBL characteristics we show are likely more related to the large-scale environment, because small-scale processes are not well constrained in MERRA-2 (e.g. Witte et al. 2021)."

**L315 – "...have a maximum area larger than 104 sq km... " – The value here is likely wrong, given that the minimum size is 0.5degx0.5deg. However, I am curious how sensitive is this size to the 0.5x0.5deg minimum threshold used to identify POCs. I suspect not so much, but I am curious, given that the results here differ from Watson-Parris et al. (2021).**

*The trajectories do follow each POC prior to them reaching the minimum size needed to be identified, therefore these results should not be sensitive to the window size.*

**L317-318 – Precipitation is discussed as if it's solely driven by boundary layer dynamics and microphysics, but certain environmental conditions do make precipitation formation more conducive, as was discussed by Wood et al. (2008). LWP, re, and N appear to be different before POCs form (L271-272). Are there POC and nonPOC differences in Fig 10 or 15 that would be conducive to increasing LWP or decreasing N?**

*We do find differences in LWP, re, and N prior to POC development (Figure R-3), however there do not appear to be any environmental differences between POC and*

*non-POC air (Figure R-5) that would be conducive to the differences shown in LWP and N.*

---

## Referee Report (RR1)

acp-2021-1001
Reviewer 1, 2nd review

The authors have addressed many of my comments on the original version of the manuscript to a large degree, but some key issues remain.

Please do a search for "CONTROL" in text and figure labeling and make sure all are changed to "CLOSED" for example, Table 2 annotation of "CONTROL" needs to be changed to "CLOSED".

a)  Related to previous comment: Further detail is needed regarding the statistics of POCs that do not reclose in order to distinguish among several scenarios. In particular, the size of the subset of POCs that essentially run off the cloud deck needs to be quantified

Since only 12 POC cases reclose out of 141, nearly all the "ends" are going off the edge of the cloud deck.  To reduce the ambiguity of what "end" and "end time" of POC means in the text, suggest replace "end" with "exit cloud deck" in text and esp. in caption for Table 1 and captions for Figures 5, 7 in the paper.

b)  Inadequate response to previous comment: B) Retrievals of cloud properties from areas of broken cloud have high uncertainties

*Authors' response:*

*We agree that the uncertainties of cloud property retrievals inside POCs may be problematic, however the primary reason we include the nighttime results - i.e. we expect that if we see similar patterns in both the daytime and nighttime results, we suspect that the changes found in the different cloud properties are real and not due to uncertainty resulting from broken cloud. To clarify this, we added lines 161-164:*

*"Overall, uncertainties in microphysical retrievals inside POCs (Coakley et al. 2005) are unavoidable. However, the nighttime retrievals provide an independent comparison to the daytime retrievals that can lend confidence to the patterns, not magnitudes, in the daytime microphysical patterns and alleviate concerns about daytime retrievals due to broken cloud (Coakley et al. 2005)."*

Effectively, this is use of bad data to check for consistency with bad data. Reviewer 2 also expressed major concerns about the nighttime retrievals of cloud properties. In the authors' response to Reviewer 2 "*Yes, LWP and N are derived at night, and, because of the*

*differences in day and night retrievals of COD and re, we find the magnitude of*

*LWP and N are smaller at night than during the day. However, we have not found*

*any studies that have evaluated the validity of deriving LWP and N from nighttime*

*retrievals."*

**Authors do not seem to understand that VIS/IR satellite retrieved cloud properties in areas of broken cloud (and especially in broken cloud at night) have such high uncertainties that they are effectively unusable for properties such cloud amounts, droplet effective radii, optical depths, cloud altitudes, cloud liquid water, and column droplet number concentrations.**

**Given the issues, suggest remove all material related to nighttime retrievals and limit all daytime retrievals to areas of high cloud fraction. This is an area where the satellite retrievals cannot stand in for in situ measurements from aircraft. The paper's credibility is substantially weakened by inclusion of such problematic data.**

 c)  Related to author's clarification of "before" and "after" periods:

*Authors' response: Regarding the "before" period - it ends 10 minutes before the*

*cloud transition, and the "after" period represents 10 minutes after POC end.*

*Lines 129-130 have been modified to clarify this:* "The trajectories are run

forward and backward from 10-minutes before POC development up to 6 hours

before POC development and 6 hours after POC dissipation."
**Figure 14-16, utilize POC characteristics "After" POC end which for 129 of the 141 POCs means the POC went off the edge of the cloud deck. Please clarify how one can obtain *any cloud characteristics* in the hours after a POC has moved off the edge of the cloud deck?**

New Comment: Please clarify: Lines 465-473: While evidence that most POCS do not reclose before moving off of the edge of the cloud deck is in the paper, for finding in paper that POCs "tend to accelerate the StCu to Cu transition and not to reclose…", *evidence specially related to accelerating is not presented, nor is the definition of acceleration made clear in this context.* Mention is made of Yamaguchi et al. (2017) LES paper but that is not evidence from *this study*.

---

## Author Response (AR3)

General Comments:
**Reviewer comments are bolded,** *our answers are italicized*, *any text in the manuscript that did not change is red*, and modified text is blue

We thank the reviewers and editor for their thorough examination of this manuscript, and we believe that we have now addressed your concerns. The primary major concerns of reviewer 1 centered on: 1) uncertainties in the retrieved cloud-properties in broken cloud regions, 2) problems with the nighttime cloud-property retrievals, and 3) our definition of the "after" period. Below we have addressed these concerns. Specifically, 1) we modified the main text to only include microphysical-properties on cloudy pixels that are connected to four other cloudy pixels (excluding corners) to limit biases due to broken clouds and found that our overall results and conclusions did not change, 2) we moved all discussion of the nighttime cloud-property results to an Appendix, and 3) we now explicitly state that the "after" period either means that open cells transition back to closed cells or start transitioning to Cu. See our detailed responses below.

Editor's Remarks:

**The reviewer/editor comments are bolded**, *my answers are unbolded and italicized*, and any unchanged text is red and changed text is blue

**Dear Dr. Smalley,**

**I now have a second round of reviews. While Reviewer 2 feels the manuscript is ready for acceptance with only minor corrections, Reviewer 1 remains concerned that a few technical aspects of the analysis have problems that were pointed out in the first round of reviews but have not been satisfactorily addressed. I suspect you can address these quite readily and hope you will seriously consider the merits of the Reviewer's points for my consideration.**

**As an editorial comment, much of the discussion and conclusions of the paper are focused on the potential role (or lack of it) that aerosols play in POCs. Given that there has been significant discussion on this point, as you summarize in the body of the document, I would like to suggest that you highlight related conclusions in the abstract.**

**Regards,**

**Tim Garrett**

Editor Comment:

We added the following sentence to the abstract (lines 14-16) that now addresses aerosols: *"Interestingly, there are no differences in reanalysis aerosol-optical depth between both sets of trajectories which may lead one to the interpretation that differences in aerosol concentrations are not influencing POC development or resulting in a large number that re-close. However, this largely depends on the reanalysis treatment of aerosol-cloud interactions and the product used in this study has no explicit handling of these important processes."*

Reviewer 1:

**The authors have addressed many of my comments on the original version of the manuscript to a large degree, but some key issues remain.**

**Please do a search for "CONTROL" in text and figure labeling and make sure all are changed to "CLOSED" for example, Table 2 annotation of "CONTROL" needs to be changed to "CLOSED".**

*We have changed "CONTROL" to "CLOSED" in the caption of Table 2.*

**a) Related to previous comment: Further detail is needed regarding the statistics of POCs that do not reclose in order to distinguish among several scenarios. In particular, the size of the subset of POCs that essentially run off the cloud deck needs to be quantified.**

*The number of POCs that do not reclose are given on lines 224-225: "Further breaking this down, 129 POCs never re-close, 12 POCs re-close, and 6 of the calculated trajectories leave their associated POC area prematurely."* and in your following statement. Statistics regarding POCs that do not re-close are listed in Table 1 (added after the last revision) and are referenced in the main text (lines 238 and 252).

**Since only 12 POC cases reclose out of 141, nearly all the "ends" are going off the edge of the cloud deck. To reduce the ambiguity of what "end" and "end time" of POC means in the text, suggest replacing "end" with "exit cloud deck" in text and esp. in caption for Table 1 and captions for Figures 5, 7 in the paper.**

*We feel the language is clear as stated in the text (abstract and section 3.1), Table 1, Figure 5, and Figure 7. For instance as stated on lines 243-245, "Why might POCs preferentially end (exit the StCu) during the day? We find that StCu area reaches a minimum around 12 local time (Figure s-1). As a result, we hypothesize that the tendency for POCs to exit the StCu during the day may simply be the result of a general*

*reduction in StCu extent during the daylit hours (Burleyson and Yuter, 2015), so that the StCu edge effectively moves towards the POC during sunlight.", the POCs that do not re-close are exiting the StCu deck. Therefore, to change "exit" to "end" in the text would be incorrect language. In the table and figure captions the POCs are referred to as ending because both the POCs that do and never re-close are grouped together.*

**b) Inadequate response to previous comment: B) Retrievals of cloud properties from areas of broken cloud have high uncertainties**

**Authors' response:**

**We agree that the uncertainties of cloud property retrievals inside POCs may be problematic, however the primary reason we include the nighttime results - i.e. we expect that if we see similar patterns in both the daytime and nighttime results, we suspect that the changes found in the different cloud properties are real and not due to uncertainty resulting from broken cloud. To clarify this, we added lines 161-164:**

**"Overall, uncertainties in microphysical retrievals inside POCs (Coakley et al. 2005) are unavoidable. However, the nighttime retrievals provide an independent comparison to the daytime retrievals that can lend confidence to the patterns, not magnitudes, in the daytime microphysical patterns and alleviate concerns about daytime retrievals due to broken cloud (Coakley et al. 2005)."**

**Effectively, this is using bad data to check for consistency with bad data. Reviewer 2 also expressed major concerns about the nighttime retrievals of cloud properties. In the authors' response to Reviewer 2 "Yes, LWP and N are derived at night, and, because of the differences in day and night retrievals of COD and re, we find the magnitude of LWP and N are smaller at night than during the day. However, we have not found any studies that have evaluated the validity of deriving LWP and N from nighttime retrievals."**

**Authors do not seem to understand that VIS/IR satellite retrieved cloud properties in areas of broken cloud (and especially in broken cloud at night) have such high uncertainties that they are effectively unusable for properties such cloud amounts, droplet effective radii, optical depths, cloud altitudes, cloud liquid water, and column droplet number concentrations.**

**Given the issues, suggest removing all material related to nighttime retrievals and limit all daytime retrievals to areas of high cloud fraction. This is an area where the satellite retrievals cannot stand in for in situ measurements from aircraft. The**

**paper's credibility is substantially weakened by inclusion of such problematic data.**

- *Broken cloud issues*
  - *We agree with the reviewer that regions of broken clouds are problematic, and we should have done a better job addressing this. Ideally, we would address this by removing partially-filled pixels, unfortunately this data is not publicly available. Therefore, we now limit our cloud-microphysical analysis to cloudy pixels that are bordered by four other cloudy pixels (excluding corners). This does not completely remove biases in broken-cloud areas, but it does limit the impact of pixels likely containing cloud edges. After doing this, we found that although the magnitudes of our microphysical statistics changes slightly, the overall results and conclusions do not change (ex. Figure R.1). Lines 171-175 explain this:* *"Prior studies have also found biases in retrieved optical properties in broken clouds due to cloud-edge effects (Coakley et al., 2005; Vant-Hull et al., 2007; Platnick et al., 2017; Zhu et al., 2018) and 3D radiative transfer artifacts (Zhang et al., 2012; Liang et al., 2015). GOES-R does not flag partially-filled pixels similar to MODIS (e.g. Jensen et al., 2008), meaning these biases likely influence our results. Therefore, we only include microphysical properties (COD, re, LWP, and N) on cloudy pixels connected to four other cloudy pixels (excluding corners), similar to MODIS (Platnick et al., 2017), to limit these biases."**, and all cloud-property figures (Figures 13, 14, S.2, and S.3) have been updated to account for this.*
- *Nighttime vs daytime*
  - *Yes, the nighttime retrieved cloud optical properties are problematic, and this is because the retrievals are limited to IR channels only. However, the nighttime retrievals used by GOES-R using only brightness temperatures are well established (i.e. Lin and Coakley 1993; Baum et al. 1994).*
  - *Given the inability of the GOES-R retrievals to capture the full range of COD at night and the reviewers concerns, we have modified both the methods (lines 160-167) to state that the main text focuses only on the daytime cloud-property results:* *"The day/night retrieval algorithms are fundamentally different. At night, COD is limited from 0 to 16 and $r_e$ is limited to 2 μm – 78 μm, whereas, during the day, COD can be retrieved from 0.25 to 158 and $r_e$ can be retrieved from 2 μm to 100 μm. The dynamic range is smaller at night because the emissivity of larger particles is similar at 11.2-μm and 12.3-μm, resulting in a smaller range of COD and $r_e$ values that can be discerned (Lin and Coakley Jr. 1993). An effect of the limited range of nighttime optical depth retrievals is that a noticeable*

*fraction of nighttime CODs are exactly 16. The diminished range of COD and $r_e$ limits our analysis of cloud-property differences between the POC and CLOSED trajectories. Therefore, we focus only on the daytime cloud property results in the main text and use the nighttime results as a comparison dataset as discussed in Appendix A.".*

○ *To indicate that we view the nighttime retrievals as less trustworthy but still complementary to the daytime retrievals, we have moved all discussion of nighttime cloud-properties to Appendix A.*

[Figure]

*Figure R.1:* The median changes in daytime cloud fraction (panel A), cloud optical depth (panel B), liquid water path (panel C), effective radius (panel D), and cloud drop number concentration (panel E) for the six hours before until ten hours after POC development are shown along all POC (red) and CLOSED (blue)

trajectories, where the red and blue fill represent changes in the interquartile range. The number of valid samples for both the GOES-16 microphysical properties (purple) and GOES-16 cloud fraction (green) are shown in panel F). Gray shading indicates the time after POC development.

**c) Related to author's clarification of "before" and "after" periods:**

**Authors' response: Regarding the "before" period - it ends 10 minutes before the cloud transition, and the "after" period represents 10 minutes after POC end. Lines 129-130 have been modified to clarify this: "The trajectories are run forward and backward from 10-minutes before POC development up to 6 hours before POC development and 6 hours after POC dissipation."**

**Figure 14-16, utilize POC characteristics "After" POC end which for 129 of the 141 POCs means the POC went off the edge of the cloud deck. Please clarify how one can obtain *any cloud characteristics* in the hours after a POC has moved off the edge of the cloud deck?**

*We modified our definition of the "before", "during", and "after" periods after the previous round of revisions, however it still appears to be inadequate. Specific to the "after" period, we note that a POC leaving the StCu deck does not imply that cloud fraction goes to zero; rather, it indicates that a transition to the Cu regime has begun. To clarify this, we modified lines 148-151 in the methods to "Specifically, we define the "before" time as the time prior to the comparison trajectory reaching the location where the POC forms, and we define the "after" time as the time after the comparison trajectory reaches the location where the POC dissipates (for the group of POCs that re-close, open cells transition back to closed cells, and for the group that never re-closes, open cells start transitioning to Cu).", and lines 323-325 in the results to "The overall differences between the POC and CLOSED trajectories do not change much in the transition from the during to after periods, likely because most POCs never re-close before transitioning from a StCu to a Cu cloud regime."*

**New Comment: Please clarify: Lines 465-473: While evidence that most POCS do not reclose before moving off of the edge of the cloud deck is in the paper, for finding in paper that POCs "tend to accelerate the StCu to Cu transition and not to reclose...", *evidence specially related to accelerating is not presented, nor is the definition of acceleration made clear in this context***

*Accelerate in this context refers to the prior cited work (lines 650-655) discussing the potential role of precipitation in the StCu to Cu transition. We agree that a term like "accelerate" should be quantified so we modified lines 421-422 to "Our finding that POCs catalyze the StCu – cumulus transition and prevent POCs from re-closing*

*suggesting that the ability of aerosol to enhance cloud albedo is highly dependent on the current state of the cloud field."*

**Mention is made of Yamaguchi et al. (2017) LES paper but that is not evidence from *this study*.**

*This Yamaguchi paper is referenced several times in that section. However, we assume that you are referring to lines 412-414:* "Furthermore, as the open-cells within POCs continue to organize and precipitation intensity increases, the StCu deck transitions to a precipitating shallow cumulus field (Yamaguchi et al. 2017).". *Considering that we do not have any evidence of this organization occurring, we removed the preceding sentence* "Considering most POCs identified in this study never re-close, our results suggest that the development of POCs driven by organizing precipitation can mediate the timing of the stratocumulus to cumulus transition." *from the manuscript, and modified the concluding sentence of this paragraph (lines 414-416):* "These results suggest a process possibly explaining why most POC cases we observe never re-close, but more observational studies tracking POCs over a longer timeframe and across more regions are needed to draw more general conclusions."

Reviewer 2:

**The authors have addressed my original comments and concerns, and I appreciate the additional analysis that examined the growth rate of POCs (Sect 3.2) and how cloud properties evolve over the course of the POC lifetime (Fig 13) .**

**I only have technical comments and two minor suggestions below to consider before publication.**

**L207 – citation needs to be outside of parentheses.**

*On line 213, "(Romps, 2017)" is now "Romps (2017)".*

**Fig9 and Fig 11a: The information in the two plots appear to be almost identical. For this reason, I wonder if one (Fig. 9) might be removed.**

*We agree that these two plots are redundant, but we think both are necessary. Specifically, we chose to keep both because we wanted to show both plots with different temporal resolutions (daily and 10-minute) to make two separate points (one demonstrating how/why we separated days with many and no POCs and the correlation between the number of POCs every 10 minutes and StCu area).*

**L382 – typo with 104 km2**

*This has been corrected to "$10^4$ $km^2$" (line 381).*

**L419-422 – Perhaps it is worth mentioning again the result in Section 3.1 that noted that while modeling studies have examined what aerosol increases are necessary to close POCs, you find no evidence for such processes occurring in your analysis.**

*We do not think that a statement regarding the ability of aerosols to potentially close POCs is appropriate here, because MERRA-2 aerosols are not fully interactive with rain, and we cannot test this with our data.*

**L426-427 – This sentence needs to be fixed: the same phrase is repeated.**

*Lines 426-428 have been modified to "This study demonstrates, most importantly, that the improved spatio-temporal resolution of the current generation of geostationary sensors and the associated data product suite provides an important tool in evaluating the temporal dimension of POCs for future studies.".*